# Crystal structures of bacterial small multidrug resistance transporter EmrE in complex with structurally diverse substrates

**Ali A Kermani[1†], Olive E Burata[1,2†], B Ben Koff[1], Akiko Koide[3,4], Shohei Koide[3,5], Randy B Stockbridge[1,6]***

[1]Department of Molecular, Cellular, and Developmental Biology, University of Michigan, Ann Arbor, United States; [2]Program in Chemical Biology, University of Michigan, Ann Arbor, United States; [3]Laura and Isaac Perlmutter Cancer Center, New York University Langone Medical Center, New York, United States; [4]Department of Medicine, New York University Grossman School of Medicine, New York, United States; [5]Department of Biochemistry and Molecular Pharmacology, New York University Grossman School of Medicine, New York, United States; [6]Program in Biophysics, University of Michigan, Ann Arbor, United States

**\*For correspondence:**
stockbr@umich.edu

[†]These authors contributed equally to this work

**Abstract** Proteins from the bacterial small multidrug resistance (SMR) family are proton-coupled exporters of diverse antiseptics and antimicrobials, including polyaromatic cations and quaternary ammonium compounds. The transport mechanism of the *Escherichia coli* transporter, EmrE, has been studied extensively, but a lack of high-resolution structural information has impeded a structural description of its molecular mechanism. Here, we apply a novel approach, multipurpose crystallization chaperones, to solve several structures of EmrE, including a 2.9 Å structure at low pH without substrate. We report five additional structures in complex with structurally diverse transported substrates, including quaternary phosphonium, quaternary ammonium, and planar polyaromatic compounds. These structures show that binding site tryptophan and glutamate residues adopt different rotamers to conform to disparate structures without requiring major rearrangements of the backbone structure. Structural and functional comparison to Gdx-Clo, an SMR protein that transports a much narrower spectrum of substrates, suggests that in EmrE, a relatively sparse hydrogen bond network among binding site residues permits increased sidechain flexibility.

## Editor's evaluation

*E. coli* EmrE and other members of the SMR family of transporters utilize the proton motive force to pump out a broad spectrum of antibiotics, thereby contributing to multi-drug resistance. Here, a new multipurpose crystallization chaperone is used to determine the structure of EmrE in apo form and in complex with various substrates. The strength of the manuscript is in the description of six new structures of EmrE at a resolution sufficient for building an atomic model and understanding how the antimicrobial agents bind, allowing robust conclusions to be drawn regarding the molecular details of binding of the antimicrobial agents. The report will be of interest to both those studying antibiotic resistance and those studying transporters.

## Introduction

The small multidrug resistance (SMR) family of microbial membrane proteins is a well-studied family composed of primitive dual-topology proton-coupled transporters. The SMR family has two major physiological subtypes that can be distinguished based on sequence (*Kermani et al., 2020*). Representatives of the 'Gdx' (guanidinium export) subtype export a bacterial metabolite, guanidinium ion (Gdm⁺), in exchange for two protons (*Kermani et al., 2018*; *Nelson et al., 2017*). Representatives of the 'Qac' (quaternary ammonium compound) subtype are proton-coupled exchangers of quaternary ammoniums and other hydrophobic, cationic compounds. Since the first quaternary ammonium antiseptics were introduced approximately one hundred years ago, proteins from the Qac cluster have been closely associated with the spread of multidrug resistance elements (*Gillings, 2017*; *Pal et al., 2015*; *Russell, 2002*; *Zhu et al., 2017*).

Many bacteria possess SMR proteins belonging to both subtypes. Transporters from the Qac and Gdx clusters do not overlap in terms of physiological role: the Qac proteins do not transport Gdm⁺ and require additional hydrophobicity in transported substrates, whereas the Gdx transporters require substrates to have a guanidinyl moiety and cannot export quaternary ammoniums or other cations (*Kermani et al., 2020*). However, the two subtypes transport an overlapping subset of hydrophobic substituted guanidinium ions and share high sequence conservation (~35% sequence identity), strongly suggesting conservation of the overall fold.

The best-studied of the Qac proteins is the *E. coli* member, EmrE. The substrate repertoire of EmrE includes planar, conjugated aromatic ring systems, quaternary ammoniums and phosphoniums (with or without aromatic substituents), and substituted guanidiniums. EmrE also provides resistance to biocides from these substrate classes with long alkyl tails, such as benzalkonium and cetyltrimethylammonium, which are found in common household antiseptics. Mechanisms to explain the transport promiscuity have been proposed, typically focusing on protein dynamics as a feature that allows it to transport many different substrates (*Jurasz et al., 2021*; *Robinson et al., 2017*). However, the structural basis for substrate binding is unknown, and for many years, structural information was limited to low-resolution models without loops or sidechains (*Fleishman et al., 2006*; *Ubarretxena-Belandia et al., 2003*), impeding a full description of the molecular mechanism. A previous crystal structure of EmrE was unreliable for molecular analysis, with no sidechains modeled, poor helical geometry, and helices too short to span the membrane (*Chen et al., 2007*). Computational models constrained by the low-resolution data have also been proposed (*Ovchinnikov et al., 2018*; *Vermaas et al., 2018*). Recently, high-resolution structural information for the SMR family has begun to emerge. First, crystal structures of a Gdx homologue from *Clostridales*, Gdx-Clo, were resolved in complex with substituted guanidinium compounds including octylguanidinium (*Kermani et al., 2020*). In addition to revealing the binding mode of the guanidinyl headgroup, the structure of Gdx-Clo with octylguanidinium showed that hydrophobic repacking of residues lining one side of the binding pocket opens a portal from the substrate binding site to the membrane interior, accommodating the substrate's long alkyl tail. In addition, a model of an EmrE mutant with reduced conformational exchange dynamics, S64V, computed from extensive NMR measurements, was also reported recently (*Shcherbakov et al., 2021*).

Here, we report several crystal structures of EmrE, including a low-pH (proton-bound) structure and five structures in complex with structurally diverse quaternary phosphonium, quaternary ammonium, and planar aromatic substrates. Structure determination was facilitated by repurposing a monobody crystallization chaperone that we originally developed for Gdx-Clo (*Kermani et al., 2020*). The EmrE structure reported here has high structural similarity to Gdx-Clo, but with notable differences in the hydrogen bond network of the substrate-binding site. The various substrates are accommodated by EmrE with minimal changes in the backbone structure. Instead, binding site tryptophan and glutamate sidechains adopt different rotamers to accommodate different drugs. These sidechain motions expand or reduce the binding pocket and provide ring-stacking interactions for structurally disparate substrates. We propose that, compared with the closely related but more selective SMR, Gdx-Clo, a reduced network of hydrogen bond interactions in the EmrE binding site allows sidechain flexibility to accommodate polyaromatics, substituted guanidinyl compounds, and quaternary ammoniums and phosphoniums without requiring substantial alteration of EmrE's backbone configuration.

## Results

### Engineering of EmrE to introduce a monobody binding site

We recently solved a crystal structure of a metabolic $Gdm^+$ exporter from the SMR family, Gdx-Clo (*Kermani et al., 2020*). For this effort, we selected monobody crystallization chaperones from large combinatorial libraries (*Koide et al., 2012*; *Sha et al., 2017*), which aided in crystallization of the transporter. Upon structure determination, we noticed that the interface between Gdx-Clo and mono-body L10 is limited to a nine-residue stretch of loop one that is relatively well-conserved among SMR proteins (*Figure 1A*). Moreover, crystal contacts are mediated almost entirely by the monobody, whereas contacts between the transporter and a symmetry mate are limited to five hydrophobic residues contributed by $TM4_A$ and $TM4_B$ (*Figure 1—figure supplement 1*). These observations suggested that conservative mutagenesis of EmrE loop one to introduce the Gdx-Clo residues might permit monobody L10 binding in order to facilitate crystallization of EmrE. We therefore designed a triple mutant, E25N, W31I, V34M, which we call $EmrE_3$. Previous studies showed minimal functional pertur-bation upon mutation of E25 and W31 to Ala or Cys (*Elbaz et al., 2005*; *Yerushalmi and Schuldiner, 2000*). All three residues are located at a distance from the substrate-binding site, and none of the three are conserved in the SMR family.

In accord with these observations, solid supported membrane (SSM) electrophysiology experi-ments showed that $EmrE_3$ mutant is active and transports representative substrates tetrapropylammo-nium ($TPA^+$) and phenylguanidinium ($PheGdm^+$). Upon perfusion with substrate, negative capacitive currents are evoked, indicating an electrogenic transport cycle, with substrate transport coupled to the antiport of ~2 $H^+$, as has been previously reported for these (*Kermani et al., 2020*) and other substrates (*Adam et al., 2007*; *Rotem and Schuldiner, 2004*; *Soskine et al., 2004*). In SSM experi-ments, the peak capacitive current corresponds to the initial rate of substrate transport (*Bazzone et al., 2017*). The SSM electrophysiology traces are very similar for WT EmrE and $EmrE_3$ (*Figure 1B*). Measurements of peak currents as a function of substrate concentration were fit to the Michaelis-Menten equation, yielding $K_m$ values within twofold of those measured for WT EmrE (*Figure 1B*, *Figure 1—figure supplement 2*). Microscale thermophoresis experiments show that $EmrE_3$ binds monobody L10 with a $K_d$ of 850 nM (*Figure 1C*, *Figure 1—figure supplement 3*), indicating that these small modifications at surface exposed residues were sufficient to create a monobody binding site. Similar to our observation for Gdx-Clo (*Kermani et al., 2020*), addition of saturating L10 monobody (10 µM) depresses transport currents mediated by $EmrE_3$ by about 40% but does not altogether inhibit substrate transport (*Figure 1D and E*). Currents are fully restored upon subsequent incubation with monobody-free solution. Thus, $EmrE_3$ is functionally equivalent to WT EmrE, is capable of binding monobody L10, and retains function when this monobody is bound.

### Structure of $EmrE_3$ without ligand at pH 5.2

When combined with monobody L10, $EmrE_3$ crystallized and diffracted to a maximum resolution of 2.9 Å. The crystallization conditions differed from those used for the Gdx-Clo/monobody complex, but the space group, C121, and approximate dimensions of the unit cell were the same (*Kermani et al., 2020*). We solved the structure using molecular replacement, with the L10 monobodies and the first three helices of each Gdx-Clo monomer as search models. After phasing, loop 3 and helix 4 were built into the experimental density followed by iterative rounds of refinement (*Figure 2A*, *Table 1*, *Figure 2—figure supplement 1A,B*). The model was validated by preparing a composite omit map in which 5% of the atoms in the model were removed at a time (*Terwilliger et al., 2008*; *Figure 2—figure supplement 1C,D*). Our $EmrE_3$ model corresponds well with the composite omit maps, suggesting that model bias introduced by using Gdx-Clo as a molecular replacement search model does not unduly influence our model of $EmrE_3$.

The structure of the $EmrE_3$/L10 complex (*Figure 2A*) shows an antiparallel $EmrE_3$ dimer bound to two monobodies in slightly different orientations via the loop one residues. The crystal packing is similar to Gdx-Clo, with the majority of contacts mediated by monobody. The introduced E25N sidechain of $EmrE_3$ is within hydrogen bonding distance of a tyrosine sidechain contributed by the monobody, and W31I contributes to a hydrophobic patch of the transporter/monobody interface. These interactions are homologous to those observed for the Gdx-Clo/L10 complex. The third mutant sidechain of $EmrE_3$, V34M, does not interact with monobody in this structure, and therefore might not be necessary for monobody binding to $EmrE_3$.

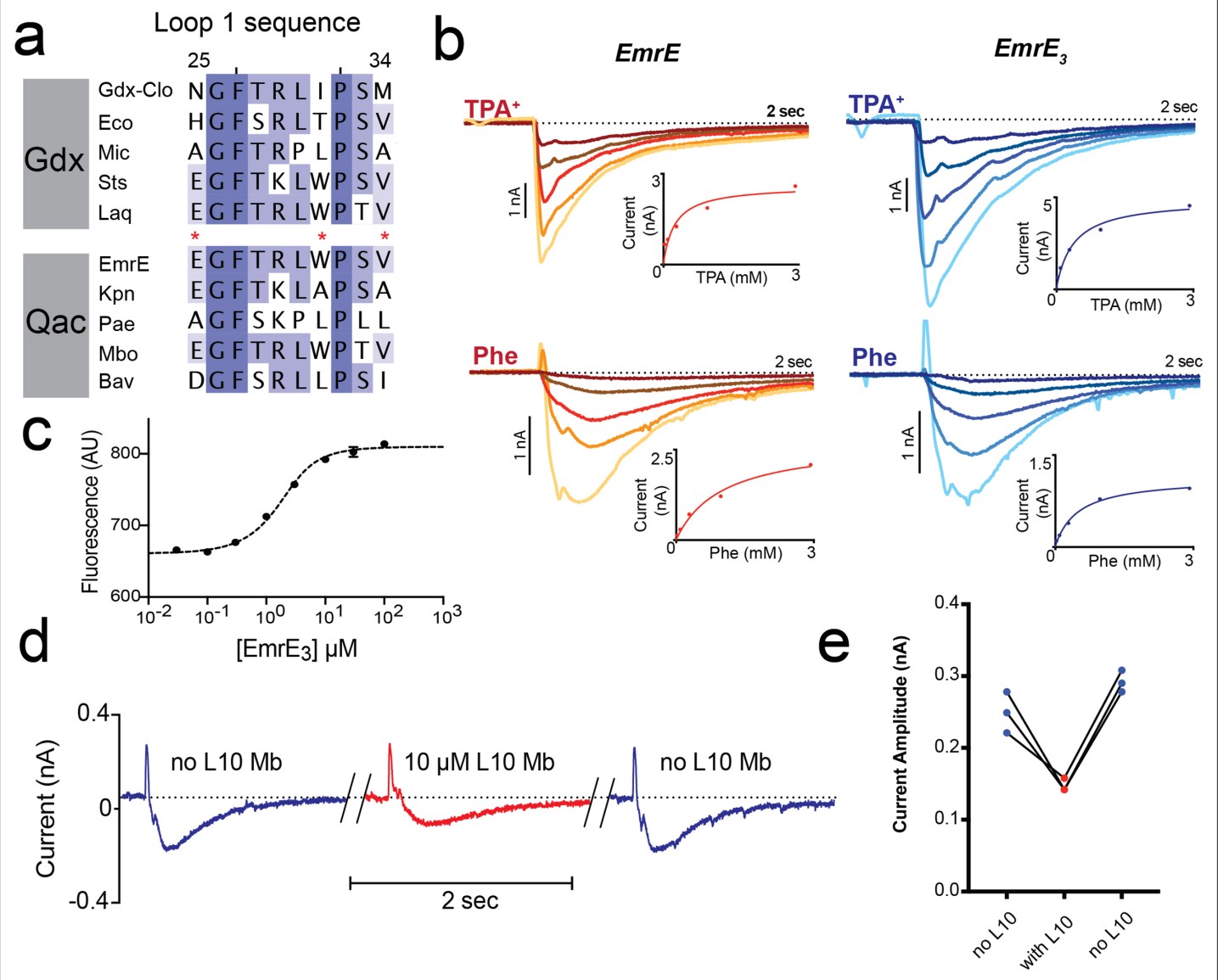

**Figure 1.** Introduction of monobody binding epitope to EmrE. (**A**) Sequence alignment for loop 1 of selected SMR proteins, numbered according to EmrE sequence. From top to bottom: representative Gdx sequences (*Clostridiales* bacterium oral taxon 876, *Escherichia coli*, *Micromonospora*, *Streptomyces tsukubensis*, and *Leifsonia aquatica*) and representative Qac sequences (*Escherichia coli*, *Klebsiella pneumoniae*, *Pseudomonas aeruginosa*, *Mycobacterium bovis*, and *Bordetella avium*). Positions mutated in the EmrE₃ construct (E25N, W31I, V34M) are indicated with red asterisks. Sequence conservation analysis for this loop is shown in *Figure 4*. (**B**) Representative currents evoked by perfusion of WT EmrE or EmrE₃ sensors (shades of red and blue, respectively) with 30 µM – 3 mM TPA⁺ (top panels) or PheGdm⁺ (Phe, lower panels). Insets show plot of peak current amplitude as a function of substrate concentration for a representative titration performed using a single sensor. Solid lines represent fit of datapoints from a single titration series to the Michaelis-Menten equation. $K_m$ values for independent replicates are reported in *Figure 1—figure supplement 2*. (**C**) Microscale thermophoresis measurement of EmrE₃ binding to monobody L10. Points and error bars represent mean and SEM of three independently prepared samples. Where not visible, error bars are smaller than the diameter of the point. Dashed line represents fit to *Equation 1* with $K_d$ = 850 nM. Representative raw data trace is shown in *Figure 1—figure supplement 3*. (**D**) EmrE₃ currents evoked by 1 mM PheGdm⁺. Sensors were incubated for 10 min in the presence (red traces) or absence (blue traces) 10 µM monobody L10 prior to initiating transport by perfusion with PheGdm⁺. Currents shown are from a representative experimental series using a single sensor preparation. (**E**) Peak currents measured for three independent perfusion series performed as in panel D. Peak currents decreased an average of 40% ± 1.5% in the presence of monobody.

The online version of this article includes the following source data and figure supplement(s) for figure 1:

**Source data 1.** SSM electrophysiology traces for EmrE and EmrE₃ with varying concentrations of TPA⁺ and phenylGdm⁺.

**Source data 2.** Peak currents for EmrE or EmrE₃ as a function of TPA⁺ or phenylGdm⁺ concentration.

**Source data 3.** Changes in MST fluorescence as a function of EmrE₃ concentration.

*Figure 1 continued*

**Source data 4.** SSM electrophysiology traces for EmrE$_3$ with and without L10 monobody addition.

**Source data 5.** Peak current values for replicate measurements of EmrE$_3$ currents in the presence and absence of monobody L10.

**Source data 6.** K$_m$ values for TPA$^+$ and phenylGdm$^+$ transport by EmrE and EmrE$_3$.

**Figure supplement 1.** Crystal lattice for Gdx-Clo/L10 monobody complex (PDB: 6WK8).

**Figure supplement 2.** K$_m$ values for TPA$^+$ and PheGdm$^+$ transport by EmrE$_3$ (blue) and WT EmrE (red).

**Figure supplement 3.** Representative microscale thermophoresis traces for monobody L10 in the presence of 30 nM – 10 μM EmrE$_3$.

In our EmrE$_3$ model, the positions of the helices agree with those observed in existing low-resolution electron microscopy maps of EmrE (*Ubarretxena-Belandia et al., 2003*; *Figure 2—figure supplement 2A*). Compared with a previous MD model based on that EM data (*Vermaas et al., 2018*), our current EmrE$_3$ crystal structure has a C$_\alpha$ RMSD of 2.5 Å, with close correspondence of residues that contribute to the substrate-binding pocket (*Figure 2—figure supplement 2B*). Although EmrE$_3$ has high structural similarity to Gdx-Clo (C$_\alpha$ RMSD 1.2 Å for the dimer), the structures display clear differences in subunit packing. Relative to Gdx-Clo, in EmrE$_3$ helices 1–3 of the A subunit, which line the binding pocket, are each displaced by 1.5–2.5 Å (*Figure 2—figure supplement 2C*). These shifts slightly expand the aqueous cavity of EmrE$_3$ relative to Gdx-Clo.

As in Gdx-Clo, the two monomers adopt different structures. Monomers A and B differ from each other in the relative orientation of their two lobes (residues 1–66 and 67–103) about a fulcrum at the conserved GVG motif in helix 3 (residues 65–67; *Figure 2B*). The angle of the bend in TM3 at the GVG sequence is somewhat more pronounced in monomer A (17°) than in monomer B (9°). The observed architecture is in accord with the proposed conformational swap of two structurally distinct monomers (*Morrison et al., 2011*).

The residue S64 is positioned immediately before the GVG fulcrum, at the boundary of lobe 1 and lobe 2 for each EmrE$_3$ subunit. In the crystal structure, the S64 sidechains contributed by the two subunits are within hydrogen bonding distance and geometry, with strong contiguous electron density between them (*Figure 2C*). Due to the antiparallel architecture, the outward- and inward-facing conformations of the transporter are expected to be structurally identical and related by twofold symmetry about an axis parallel to the plane of the membrane (*Fleishman et al., 2006*). Thus, the S64 interaction should be preserved when the transporter is open to the opposite side of the membrane; we therefore imagine that the S64 sidechains remain hydrogen bonded to each other during the entire transport cycle, forming the pivot point around which the conformational change occurs.

In the absence of ligand, EmrE$_3$ possesses a deep, spacious aqueous pocket that is accessible from one side of the membrane (*Figure 2A*). The E14 sidechains contributed by both subunits define the edges of this binding pocket. E14 is invariant in the SMR family and essential for binding both substrate and protons (*Yerushalmi and Schuldiner, 2000*). The present crystals formed at pH 5.2, at which both E14 sidechains are expected to be protonated (*Li et al., 2021*; *Morrison et al., 2015*). There is a small, spherical density in the vestibule between W63$_B$ and E14$_A$ that is consistent with a water molecule, although no other ordered water molecules are visible at this resolution (*Figure 2—figure supplement 3*). The cross-subunit interaction between Y60$_B$ and E14$_A$ proposed by Vermaas et al. is observed (*Figure 2D*). A conserved hydrogen bond acceptor, T18$_A$, is located one helical turn down from E14$_A$ and engaged in an intrasubunit interaction with Y40$_A$ (*Figure 2D*).

As in Gdx-Clo, the TM2 helices splay apart on the open side of the transporter, defining a portal from the membrane to the substrate binding site that is lined with hydrophobic sidechains (*Figure 2E*). This portal may play a dual role, rearranging to allow alkyl substituents to reside in the membrane during the transport cycle, as well as providing the opportunity for hydrophobic drugs to diffuse laterally from the membrane into the substrate binding site. Aromatic residues contributed by loop 1$_A$, including the highly conserved F27 sidechain, are wedged between the hydrophobic sidechains lining helices 2$_A$ and 2$_B$, sealing the closed side of the transporter (*Figure 2E*).

## Structures of substrate-bound EmrE$_3$

To understand how different substrates interact with EmrE, we screened a variety of transported compounds in crystallization trials. We were able to obtain diffracting crystals in the presence of five structurally diverse compounds transported by EmrE: monovalent planar aromatic harmane (3.8 Å),

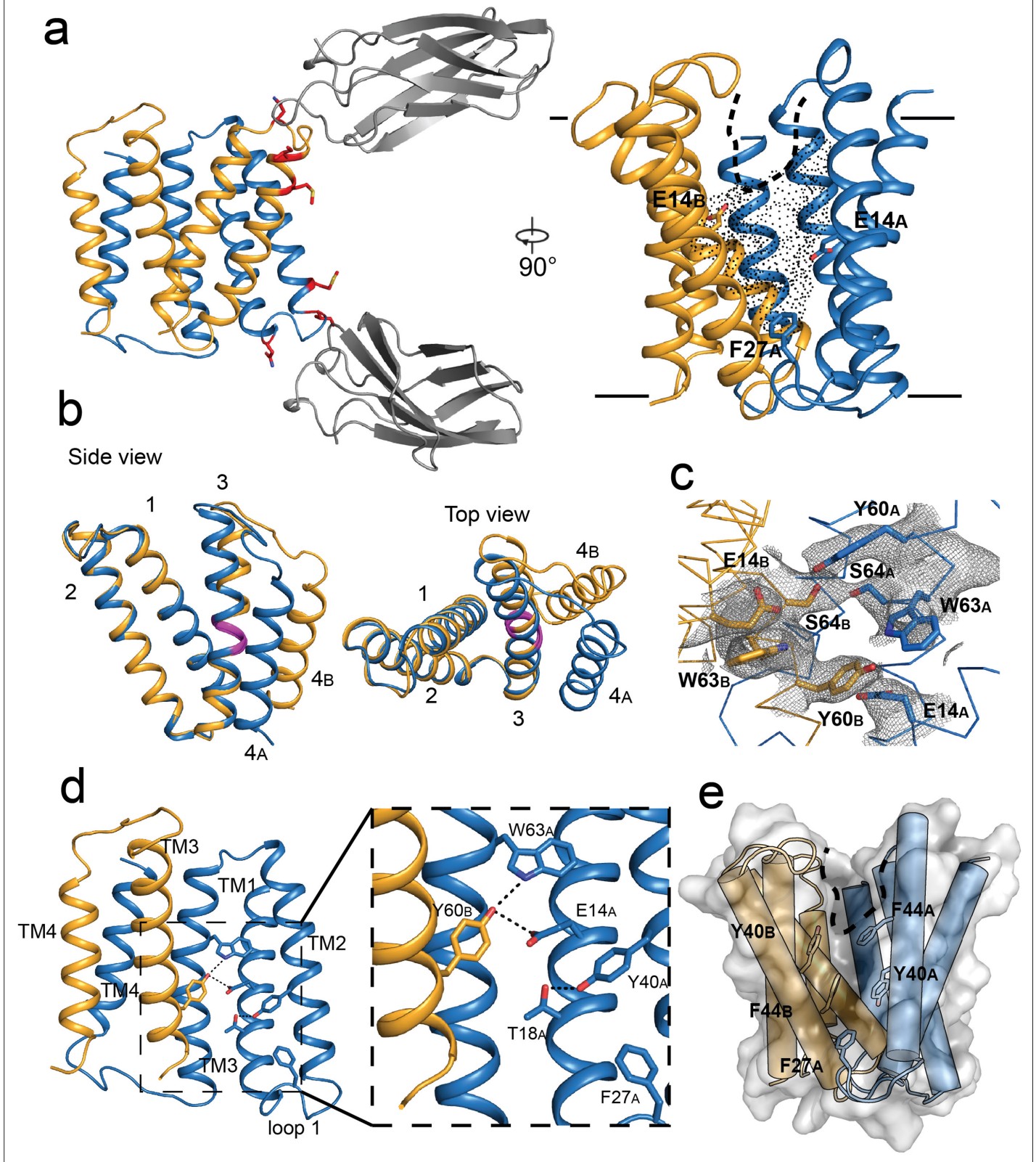

**Figure 2.** Crystal structure of EmrE₃. (**A**) Subunits A and B are shown in blue and orange, respectively, and monobody L10 is shown in gray. In the left panel, mutated residues E25N, W31I, V34M are shown in red with sidechain sticks. In the right panel, the monobodies are removed for clarity. E14$_A$, E14$_B$, and F27$_A$ are shown as sticks, and the aqueous accessible region of the transporter is indicated with dots. Approximate membrane boundaries are shown as solid lines, and the boundary of the membrane portal is shown as a dashed line. (**B**) A (blue) and B (orange) subunits of EmrE₃, aligned

*Figure 2 continued on next page*

*Figure 2 continued*

over residues 1–63. The GVG fulcrum sequence in TM3 is colored in magenta. (**C**) S64 and surrounding sidechains with $2F_o$-$F_c$ density shown as gray mesh (contoured at 1.0 σ within 2 Å of selected residues). (**D**) Y60$_B$ hydrogen bonding network. EmrE dimers are shown with TM1 and TM2 of subunit B (orange) removed for clarity. Lower panels show zoomed in view. In each view, interactions within hydrogen bonding distance and geometry are shown as dashed lines. E. Surface rendering of EmrE$_3$. TM2 sidechains that line the portal are shown as sticks.

The online version of this article includes the following figure supplement(s) for figure 2:

**Figure supplement 1.** EmrE$_3$ maps.

**Figure supplement 2.** Structural comparison of EmrE$_3$ crystal structure with electron microscopy maps, theoretical model, and Gdx-Clo.

**Figure supplement 3.** Sidechain density in the EmrE$_3$ binding site.

divalent planar aromatic methyl viologen (3.1 Å), quaternary phosphoniums tetraphenylphosphonium (TPP$^+$; 3.4 Å) and methyltriphenylphosphonium (MeTPP$^+$; 3.2 Å), and quaternary ammonium benzyltrimethylammonium (3.9 Å) (*Table 1*). We were unable to generate crystals that diffracted to high resolution in the presence of metformin, benzalkonium, cetyltrimethylammonium, or octylguanidinium. Phases of the EmrE$_3$/substrate/L10 monobody complexes were determined using molecular replacement with the pH 5.2 structure as a search model. Although the crystallization conditions varied for each substrate, the TPP$^+$-, MeTPP$^+$-, benzyltrimethylammonium-, and harmane-bound proteins crystallized in the same unit cell as proton-bound EmrE$_3$, with one copy of the EmrE$_3$/L10 complex in the asymmetric unit. The methyl viologen-bound protein crystallized in P1 with two pseudosymmetric copies of the EmrE3/L10 complex in the asymmetric unit, organized in the same relative orientation as individual complexes in the C121 crystal form.

Since Gdx-Clo and EmrE$_3$ were both accommodated in this crystal lattice despite differences in the tilt and packing of helices 1, 2, and 3, we expect that small 1–2 Å substrate-dependent movements in the backbone of EmrE$_3$ would also be tolerated within this crystal lattice. However, in all four substrate-bound structures, the transmembrane helices and loops 1 and 2 conform almost perfectly to the pH 5.2 structure (C$_α$ RMSD = 0.5–0.65 Å), suggesting that the observed backbone conformation is the lowest energy state for both the substrate- and proton-bound transporter. Loop three is poorly ordered and adopts a different conformation in each of the structures in which it is resolved well enough to model.

For all substrate-bound structures, the maps show positive densities between the substrate-binding E14 residues, including a four-lobed density for TPP$^+$, a three-lobed density for MeTPP$^+$, and oblong densities for the harmane and the methyl viologen structures. We modeled the corresponding substrates into each of these densities (*Figure 3A and B*). All five drugs are bound at the bottom of the aqueous cavity, in overlapping positions at the midpoint of the membrane. In the two copies of the methyl viologen-bound transporter, the drug is bound in different (but overlapping) positions (*Figure 3A*, *Figure 3—figure supplement 1*). For all substrates, the center of mass is poised midway between the E14 residues. To different extents, the substrates also interact with the protein's aromatic residues via ring stacking, especially Y60 and W63.

Comparison of these structures permitted evaluation of the specific orientations of the sidechains that line the substrate binding site (*Figure 3C*). The harmane- and benzyltrimethylammonium-bound structure was excluded from this analysis because, at 3.8–3.9 Å resolution, we were not as confident about interpreting subtle changes in sidechain orientation. For the other substrates (methyl viologen, TPP$^+$, and MeTPP$^+$), this comparison showed that binding site sidechains, especially E14 and W63, adopt different rotamers, thus accommodating the differently sized substrates. For example, the carboxylate of E14$_B$ is displaced by 2.5 Å when the bulky quaternary phosphonium TPP$^+$ is bound, compared to its position when the planar methyl viologen occupies the binding site. Likewise, the position of the W63$_A$ indole ring rotates over approximately 80° depending on the substrate that occupies the binding site. To validate these observations, we performed refinements with models in which the position of the W63$_A$ or E14$_B$ sidechain was adjusted to match its position in the presence of a dissimilar substrate; the resulting difference density demonstrates that these substrate-dependent changes in sidechain rotamer are not due to model bias during the refinement (*Figure 3—figure supplement 2*, *Figure 3—figure supplement 3*). Thus, these structures provide a first suggestion of how rotameric movements of EmrE's charged and aromatic sidechains can change the dimensions of the binding pocket and interact favorably with diverse substrates.

**Table 1.** Data collection, phasing, and refinement statistics for EmrE and Gdx-Clo complexes.

| | EmrE₃/L10/MeTPP⁺ | EmrE₃/L10/TPP⁺ | EmrE₃/L10/harmane | EmrE₃/L10/methyl viologen | EmrE₃/L10, pH 5.2 | Gdx-Clo/L10, pH 5.0 | EmrE₃/L10/BM₃A⁺ |
|---|---|---|---|---|---|---|---|
| Crystallization conditions | 0.1 M LiNO₃, 0.1 M ADA pH 6.5, 32.8% PEG 600 | 0.1 M (NH₄)₂SO₄, 0.1 M HEPES pH 7.25, 30.8% PEG 600 | 0.1 M (NH₄)₂SO₄, 0.1 M HEPES pH 7.1, 33.8% PEG 600 | 0.1 M (NH₄)₂SO₄, 0.1 M ADA pH 6.3, 34.8% PEG 600 | 0.2 M NaCl, 0.1 M sodium cacodylate pH 5.2, 34% PEG 600 | 0.1 M calcium acetate, 0.1 M sodium acetate pH 5.0, 40% PEG 600 | 0.1 M NH₄SO₄, 0.1 M HEPES pH 7.25, 33% PEG 600 |
| **Data collection** | | | | | | | |
| Space group | C121 | C121 | C121 | P1 | C121 | P1 | C121 |
| Cell dimensions | | | | | | | |
| $a$, $b$, $c$ (Å) | 141.17, 50.87, 110.79 | 140.71, 50.14, 110.28 | 145.7, 51.83, 114.95 | 50.91, 75.07, 111.43 | 140.64, 49.85, 109.83 | 49.70, 74.32, 107.43 | 140.18, 50.12, 110.73 |
| α,β,γ (Å) | 90, 92.69, 90 | 90, 93.45, 90 | 90, 92.67, 90 | 92.03, 90.33, 109.20 | 90, 93.75, 90 | 93.56, 89.71, 109.92 | 90, 92.79, 90 |
| Resolution (Å) | 70.5–3.22 (3.42–3.22) | 70.2–3.36 (3.62–3.36) | 114.8–3.75 (4.37–3.75) | 70.8–3.13 (3.41–3.13) | 70.2–2.85 (3.16–2.85) | 107.2–2.32 (2.67–2.32) | 70.50–3.22 (3.42–3.22) |
| Ellipsoidal Resolution Limit (best/worst)* | 3.22/4.33 | 3.36/5.1 | 3.75/6.34 | 3.13/4.50 | 2.85/3.72 | 2.32/3.55 | 3.22/4.33 |
| % Spherical Data Completeness* | 69.0 (20.9) | 54.5 (13.6) | 44.0 (10.1) | 52.0 (11.1) | 62.0 (12.0) | 41.9 (6.0) | 69.0 (20.9) |
| % Ellipsoidal Data Completeness* | 88.6 (80.1) | 84.1 (78.5) | 82.7 (65.7) | 82.0 (72.3) | 87.0 (62.6) | 80.3 (45.6) | 88.6 (80.1) |
| $R_{merge}$* | 0.152 (0.656) | 0.349 (1.053) | 0.365 (0.752) | 0.123 (0.697) | 0.118 (1.85) | 0.089 (0.4) | 0.152 (0.656) |
| $R_{meas}$* | 0.166 (0.707) | 0.384 (1.15) | 0.396 (0.817) | 0.144 (0.814) | 0.129 (1.99) | 0.104 (0.465) | 0.166 (0.707) |
| $CC_{1/2}$ | 0.967 (0.861) | 0.779 (0.610) | 0.992 (0.862) | 0.939 (0.629) | 0.994 (0.366) | | 0.967 (0.861) |
| Mn $I$ / σ$I$* | 10.4 (2.7) | 4.0 (1.8) | 7.4 (2.3) | 7.7 (1.5) | 9.5 (1.2) | 6.5 (2.8) | 10.4 (2.7) |
| Multiplicity* | 6.6 (7.1) | 5.9 (6.2) | 6.7 (6.6) | 3.7 (3.8) | 6.4 (7.1) | 3.8 (3.8) | 6.6 (7.1) |
| **Refinement** | | | | | | | |
| Resolution (Å) | 55.3–3.22 | 55.0–3.36 | 60.2–3.91 | 32.9–3.13 | 35.2–2.85 | 35.5–2.32 | 55.3–3.91 |
| No. reflections | 8,025 | 6,097 | 3,347 | 14,194 | 11,149 | 26,026 | 5,040 |
| $R_{work}$ / $R_{free}$ | 29.4 / 33.4 | 29.0/31.4 | 34.2/34.4 | 30.0/33.1 | 30.7/32.7 | 25.1/29.5 | 33.0/36.7 |
| Ramachandran Favored | 89.4 | 89.6 | 90.9 | 89.1 | 91.0 | 92.9 | 88.7 |
| Ramachandran Outliers | 1.9 | 1.9 | 2.4 | 2.6 | 1.9 | 1.5 | 2.7 |
| Clashscore | 11.8 | 13.6 | 8.4 | 16.8 | 8.6 | 10.4 | 14.9 |
| R.m.s. deviations | | | | | | | |
| Bond lengths (Å) | 0.003 | 0.003 | 0.003 | 0.004 | 0.003 | 0.004 | 0.004 |
| Bond angles (°) | 0.70 | 0.68 | 0.60 | 0.82 | .65 | 0.70 | 0.67 |
| Coordinates in Protein Databank | 7SSU | 7SV9 | 7SVX | 7 MGX | 7MH6 | 7SZT | 7 T00 |

*Where applicable, values reported are for anisotropically truncated data performed using the Staraniso webserver (Global Phasing). See *Methods* for details.

## Structure of Gdx-Clo at pH 5 and comparison to the substrate binding site of EmrE

The overall fold and many of the binding site sidechains are shared between EmrE and Gdx-Clo, yet the two proteins have markedly different substrate selectivity profiles. We therefore sought to analyze how molecular interactions among binding site residues might explain the different substrate selectivity for EmrE and Gdx-Clo. Previous structures of Gdx-Clo were solved at pH ≥7.5 in complex

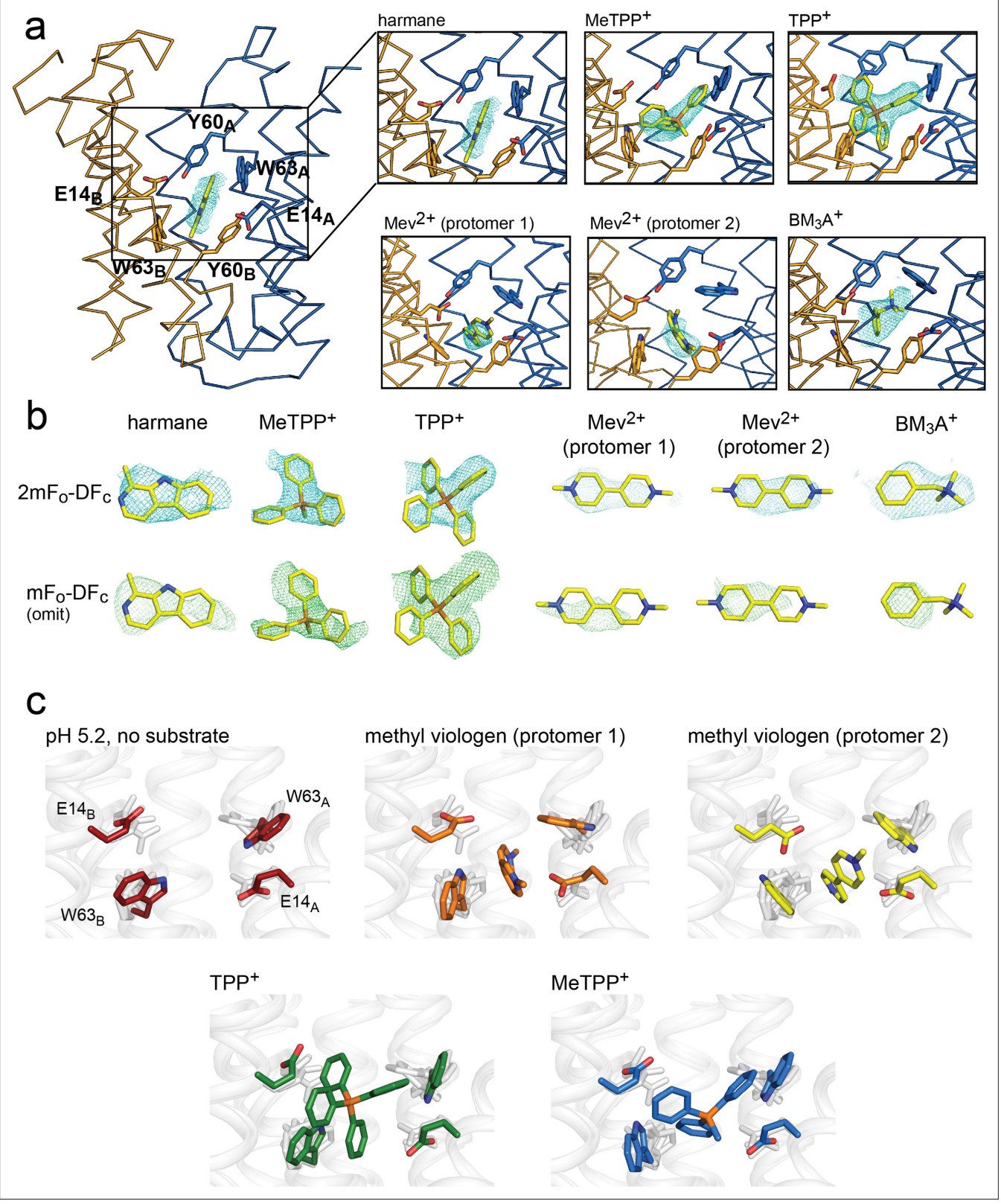

**Figure 3.** Substrate binding to EmrE₃. (**A**) Structures are shown in ribbon representation, with sidechains E14, W63, and Y60 shown as sticks. All panels are zoomed and oriented the same. $2mF_o$-$DF_c$ maps (carved 2 Å around each substrate) are shown as cyan mesh. Maps are contoured at 1σ for harmane and 1.2σ for MeTPP⁺, TPP⁺, methylviologen, and benzyltrimethylammonium (BM₃A⁺). (**B**) Top row: Substrate structures and $2mF_o$-$DF_c$ maps from the panels in A, individually rotated to view each substrate. Bottom row: $mF_o$-$DF_c$ substrate omit maps shown as green mesh. Omit maps are contoured at

*Figure 3 continued on next page*

Figure 3 continued

1.8σ for harmane and 2σ for MeTPP⁺, TPP⁺, methylviologen, and BM₃A⁺. (**C**) Comparison of E14 and W63 positions in each substrate-bound structure. Individual panels show substrate, E14, and W63 from indicated structure in color aligned with the other four structures, which are rendered in light gray.

The online version of this article includes the following figure supplement(s) for figure 3:

**Figure supplement 1.** Electron density maps of methyl viologen in different EmrE₃ protomers in the asymmetric unit.

**Figure supplement 2.** Electron density maps for W63$_A$ modeled in different positions.

**Figure supplement 3.** Electron density maps for E14$_B$ modeled in different positions.

with substituted guanidinyl compounds (**Kermani et al., 2020**). In order to compare the substrate-binding sites of Gdx-Clo and EmrE₃ in equivalent states, we solved a new structure of Gdx-Clo at pH 5.0, which is close to the value for the present low pH EmrE₃ structure, pH 5.2 (**Table 1**, **Figure 4—figure supplement 1A**). Both transporters are likely proton-bound at this pH, minimizing differences in sidechain positioning that might stem from interactions with bound substrate. This new structure of proton-bound Gdx-Clo, which is resolved to 2.3 Å, is highly similar to the structure of substrate-bound Gdx-Clo (PDB: 6WK8), with only a local change in the rotamer of the substrate-binding glutamate E13$_B$ (**Figure 4—figure supplement 1B**).

A comparison of the low-pH EmrE₃ and Gdx-Clo structures reveals conspicuous differences in the hydrogen bond network within the binding cavity (**Figure 4A and B**), despite the conservation of many key residues. In Gdx-Clo, Ser42 participates in the stack of alternating hydrogen bond donors and acceptors (W16$_{Clo}$/E13$_{Clo}$/S42$_{Clo}$/W62$_{Clo}$) that fixes the position of the central Glu, E13. Although the analogous serine (S43$_{EmrE}$) is present in EmrE, it is not playing an analogous role. A 1.5 Å displacement in helix two has distanced this Ser from the other sidechains in the binding pocket, beyond hydrogen bonding distance with W63$_{EmrE}$. Instead, S43$_{EmrE}$ is rotated away from the aqueous cavity and the central E14$_{EmrE}$ residues. Despite strict conservation of this serine among the Gdx subtype, mutation to alanine occurs in ~30% of homodimeric Qacs (**Figure 4—figure supplement 2**). In lieu of an interaction with S43$_{EmrE}$, both W63$_{EmrE}$ sidechains in EmrE adopt different rotamers compared to their counterparts in Gdx-Clo. W63$_{A, EmrE}$ is oriented so that its indole NH is within H-bonding distance of Y60$_{B, EmrE}$, although the angle between the H-bond donor and acceptor is ~30° off normal.

The fourth residue from Gdx-Clo's H-bond stack, W16$_{Clo}$, is universally conserved in Gdx proteins, but replaced with a glycine or alanine in the Qacs (G17 in EmrE). There is no equivalent H-bond donor to the central Glu in EmrE. Instead, the sidechain Y40$_{EmrE}$ occupies this space, but interacts with T18$_{EmrE}$ located one helical turn away from E14$_{EmrE}$. This pair, Y40$_{EmrE}$ and T18$_{EmrE}$, are highly conserved among the Qacs, and variable and typically hydrophobic in Gdx proteins. In Gdx-Clo, the corresponding positions are M39$_{Clo}$ and A17$_{Clo}$. This trio of correlated positions (W16$_{Clo}$/G17$_{EmrE}$, A17$_{Clo}$/T18$_{EmrE}$, and M39$_{Clo}$/Y40$_{EmrE}$) in the substrate-binding site are among the main features that differentiate the Gdx and Qac subtypes in sequence alignments (**Figure 4C**).

Y60$_{A, EmrE}$ also adopts a different orientation in EmrE relative to the position of the analogous Tyr, Y59$_{Clo}$ in Gdx-Clo. Rather than extending out of the binding pocket toward the exterior solution, as it does in Gdx-Clo, Y60$_{A, EmrE}$ is pointed down toward the S64$_{EmrE}$ diad. This rotamer would not be possible in Gdx-Clo, since this space is occupied by K101$_{Clo}$ instead, which extends from the C-terminal end of helix 4 and points down into the substrate-binding pocket toward the glutamates. K101$_{Clo}$ is completely conserved in the Gdx subtype.

The overall picture that emerges from this comparison of the Gdx-Clo and EmrE structures is that the two proteins share many binding site residues but differ in the relative organization of these residues. In Gdx-Clo, E13$_{Clo}$, S42$_{Clo}$, Y59$_{Clo}$, and W62$_{Clo}$ are constrained in a highly organized H-bond network. In EmrE, residues peripheral to the binding site have encroached on these positions, disrupting the network and reducing the number of protein hydrogen bond partners for each of these conserved sidechains.

## EmrE is tolerant of mutations that eliminate hydrogen bonding in the binding pocket

Based on structural comparison of the Gdx-Clo and EmrE-binding pockets, we hypothesize that even for conserved residues in the binding pocket, the importance of hydrogen bonding is diminished in EmrE relative to Gdx-Clo. To probe this, we performed a head-to-head comparison of SSM currents

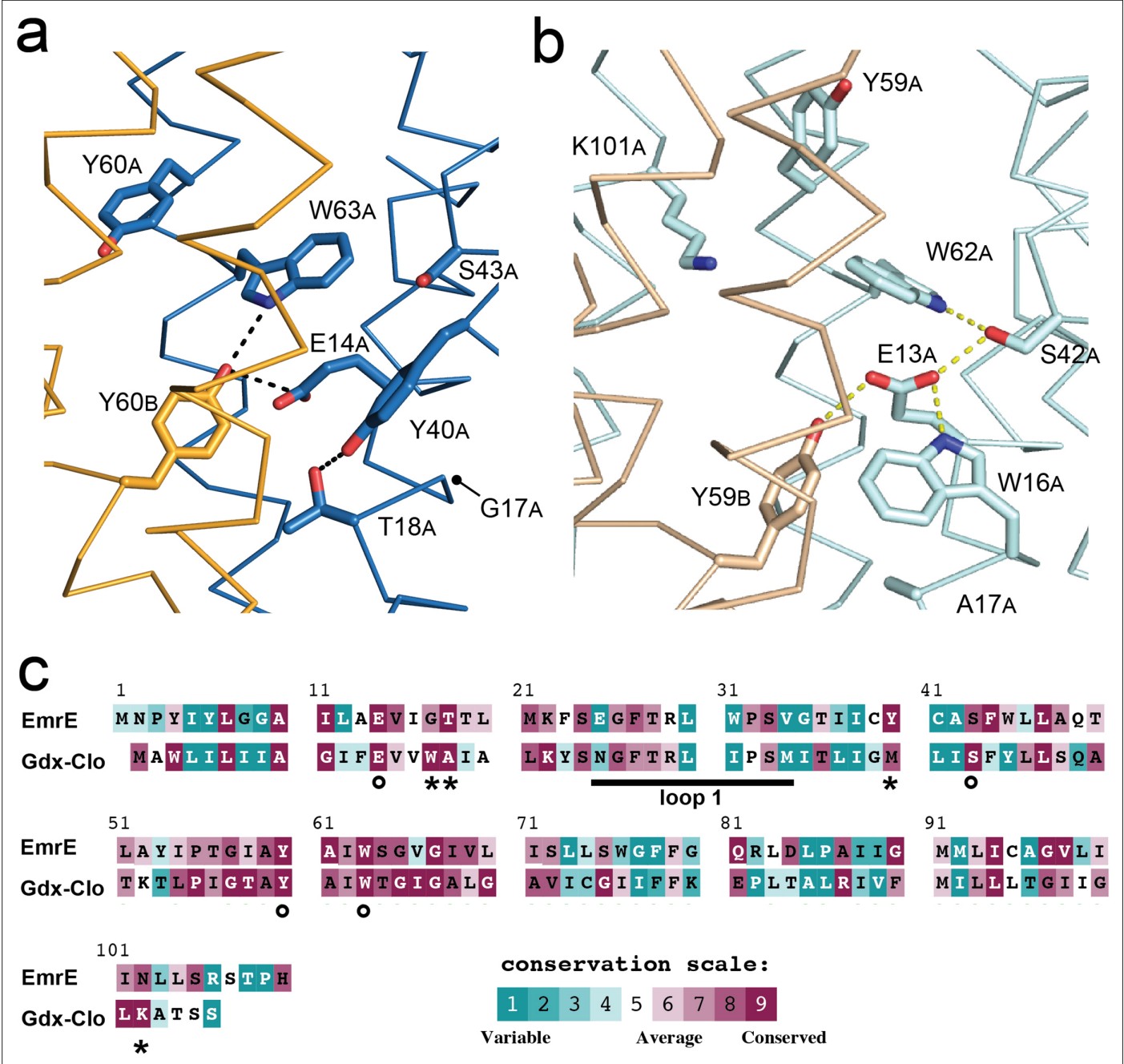

**Figure 4.** Structure and sequence conservation of substrate binding site residues in Qac and Gdx subtypes. (**A**) Substrate-binding site in EmrE, with subunit B in orange and subunit A in blue. (**B**) Substrate-binding site in Gdx-Clo, with subunit B in wheat and subunit A in pale cyan (PDB: 6WK8). For panels A and B, the proteins are shown in the same orientation. Note that residue numbering is offset by one in Gdx-Clo. Potential hydrogen bonds are shown as dashed lines. (**C**) Amino acid conservation analysis for the Qac and Gdx subtypes overlaid on exemplar sequences of EmrE and Gdx-Clo, respectively. Analysis was performed using ConSurf (*Ashkenazy et al., 2016*; *Berezin et al., 2004*). Residues that contribute to the binding pocket and that are conserved between the Qac and Gdx subtypes are indicated with an astericks. Residues that contribute to the binding pocket and that differ between the Qac and Gdx subtypes are indicated with a circle. The monobody binding loop 1 is indicated by the sold line. Alignments of representative sequences are shown in *Figure 4—figure supplement 2*.

The online version of this article includes the following figure supplement(s) for figure 4:

**Figure supplement 1.** Gdx-Clo and EmrE substrate binding sites.

**Figure supplement 2.** Sequence alignments of five representative Gdx proteins (from top to bottom: *Clostridiales* bacterium oral taxon 876, *E. coli*, *Micromonospora*, *Streptomyces tsukubensis*, and *Leifsonia aquatica*) and five representative Qac proteins (from top to bottom: *E. coli*, *Klebsiella pneumoniae*, *Pseudomonas aeruginosa*, *Mycobacterium bovis*, and *Bordetella avium*).

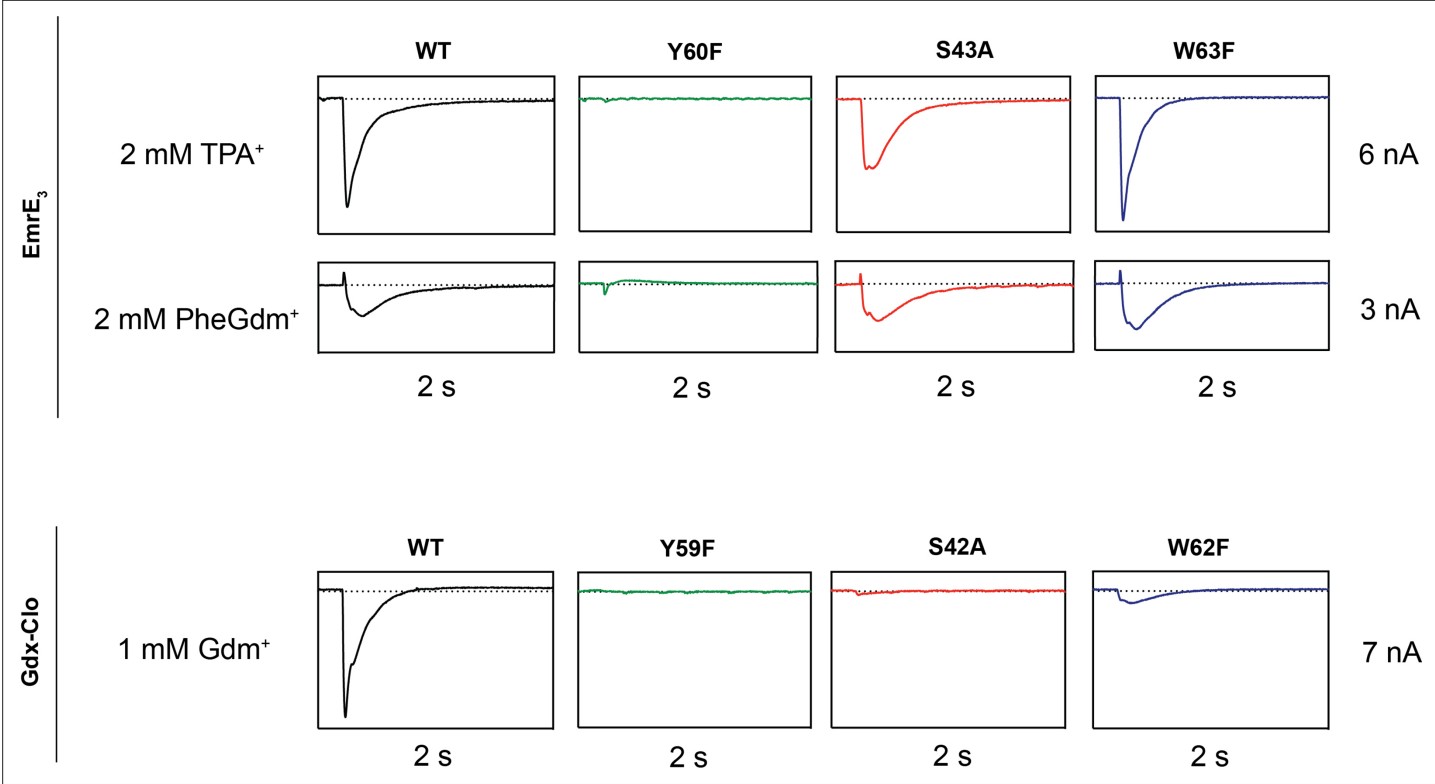

**Figure 5.** Representative SSM electrophysiology recordings for EmrE$_3$ and Gdx-Clo mutants. For EmrE$_3$, PheGdm$^+$ and TPA$^+$ traces are from the same sensor and shown on the same scale. Vertical box edges are 3 nA for PheGdm$^+$ traces, and 6 nA for TPA$^+$ traces. For Gdx-Clo, vertical box edges are 7 nA. Horizontal box edges are 2 s for all traces. Dashed line represents the zero-current level. Traces are representative of currents from three independently prepared sensors and two independent biochemical preparations. Peak current values for all replicates are reported in **Table 2**. Note that because there is some sensor-to-sensor variation in liposome fusion, comparisons of current amplitude among the mutants are qualitative.

The online version of this article includes the following source data for figure 5:

**Source data 1.** SSM electrophysiology traces for EmrE$_3$ mutants and Gdx-Clo mutants.

mediated by EmrE and Gdx-Clo proteins with mutations at three conserved positions adjacent to the functionally essential central Glu: Y59F$_{Clo}$/Y60F$_{EmrE}$, S42A$_{Clo}$/S43A$_{EmrE}$, and W62F$_{Clo}$/W63F$_{EmrE}$ (**Figure 5**, **Table 2**). All six mutant transporters were expressed at near-WT levels and monodisperse by size exclusion chromatography. For EmrE mutants, we tested transport of 2 mM PheGdm$^+$ or 2 mM TPA$^+$, and for Gdx-Clo, we tested transport of its native substrate, 1 mM Gdm$^+$. For all experiments, substrate concentration was ~4 fold higher than the transport K$_m$.

In line with its proposed role as a conformational switch (**Kermani et al., 2020**; **Vermaas et al., 2018**), no currents were observed when the binding site Tyr (Y59$_{Clo}$/Y60$_{EmrE}$) was mutated in either protein. This result recapitulates results from prior radioactive uptake studies of both mutants (**Kermani et al., 2020**; **Rotem et al., 2006**). It also establishes a dead-transporter control for our SSM electrophysiology assays. We likewise find that Gdx-Clo does not tolerate perturbation to its hydrogen bond stack. Although neither S42A$_{Clo}$ nor W62F$_{Clo}$ directly bind Gdm$^+$, both mutations eliminate Gdm$^+$ currents in SSM electrophysiology assays. In contrast, EmrE$_3$ was relatively indifferent to the S43A$_{EmrE}$ and W63F$_{EmrE}$ mutations, with robust currents evoked by both TPA$^+$ and PheGdm$^+$.

This result for S43$_{EmrE}$ reinforces the structural suggestion that the serine's functional role in the Gdx transporters is not conserved in the Qac subtype, and is also in agreement with prior transport and resistance assays that showed that S43$_{EmrE}$ modulates substrate specificity in EmrE, but is not required for transport function (**Brill et al., 2015**; **Wu et al., 2019**). The observation of robust transport by W63F$_{EmrE}$ is more surprising, since this mutant has been shown to reduce TPP$^+$ binding by two orders of magnitude, and abolish methyl viologen transport and bacterial resistance to TPP$^+$, methyl viologen, and acriflavine (**Elbaz et al., 2005**). Other mutations to W63 (to C, A, or V) also fail to provide resistance against polyaromatic substrates (**Amadi et al., 2010**; **Elbaz et al., 2005**; **Wu et al., 2019**).

**Table 2.** SSM electrophysiology peak currents (nA) for EmrE$_3$ and Gdx-Clo mutants summarized by experimental replicate.

EmrE$_3$

| | Prep 1/Sensor 1 | | Prep 1/Sensor 2 | | Prep 2/Sensor 1 | |
|---|---|---|---|---|---|---|
| | TPA$^+$ | PheGdm$^+$ | TPA$^+$ | PheGdm$^+$ | TPA$^+$ | PheGdm$^+$ |
| No protein | 0 | 0 | 0 | 0 | 0 | 0 |
| WT | −4.8 | −1.4 | −4.1 | −1.2 | −3.9 | −1.0 |
| Y60F | −0.14 | −0.03 | −0.05 | −0.07 | −0.04 | −0.05 |
| S43A | −3.7 | −1.6 | −3.9 | −1.3 | −3.2 | −1.2 |
| W63F | −5.4 | −2.0 | −4.6 | −1.5 | −4.0 | −1.0 |

Gdx-Clo

| | Prep 1/Sensor 1 (Gdm$^+$) | Prep 1/Sensor 2 (Gdm$^+$) | Prep 2/Sensor 1 (Gdm$^+$) |
|---|---|---|---|
| No protein | 0 | 0 | 0 |
| WT | −6.3 | −6.7 | −6.3 |
| Y60F | 0.04 | 0.007 | 0.6 |
| S43A | −0.15 | −0.15 | −0.15 |
| W63F | −0.60 | −0.33 | −0.30 |

To our knowledge, the consequences of W63$_{EmrE}$ mutation have not been previously investigated for non-aromatic substrates in biochemical assays. Our SSM electrophysiology results suggest that maintaining a hydrogen bond donor at W63$_{EmrE}$ is not essential, and that the conservation of W63$_{EmrE}$ is not a mechanistic requirement for EmrE transport, but is instead a determinant of aromatic substrate specificity. In agreement with this interpretation, bacterial growth assays have shown that W63$_{EmrE}$ mutants retain resistance to non-aromatic biocides (*Saleh et al., 2018*).

## Discussion

In this work, we describe substrate- and proton-bound crystal structures of the *E. coli* SMR transporter EmrE, which is wildtype except for three functionally neutral mutations that enable monobody binding, and thus, crystallization. Functional assays show that the engineered protein, EmrE$_3$ behaves like wildtype, and that the transporter remains functional in the presence of monobody. Below, we discuss the crystallization strategy, we evaluate differences between our crystal structures and a recent NMR-derived model of EmrE (*Shcherbakov et al., 2021*), and discuss the implications of our structures for understanding substrate polyspecificity by EmrE.

### The application of multipurpose chaperones for crystallization

The minimal monobody binding interface permitted a crystallization chaperone developed for Gdx-Clo to be repurposed for binding and crystallization of a new target with structural homology, but only 35% sequence identity to the original, streamlining the structural characterization process. Given the similarity of this loop among diverse SMR proteins, we think that this approach would likely facilitate the structural characterization of any target within the SMR family. Such general adapters and chaperones to facilitate structural biology have been described before for various targets (*Dutka et al., 2019*; *Koldobskaya et al., 2011*; *McIlwain et al., 2021*; *Mukherjee et al., 2020*). Although identification of a general SMR monobody was not the original intent of the monobody selection, in cases where multiple homologous targets have been identified, variants with identical or near-identical epitopes could be generated, and binders with broad utility could presumably be selected for. Especially in the case of bacterial proteins, in which there are many clinically relevant homologues from many diverse species, such general structural biology approaches hold particular promise to facilitate molecular characterization of membrane protein targets.

The monobody chaperones mediate most of the crystal contacts, permitting Gdx-Clo and EmrE to crystallize in a nearly identical unit cell, despite some structural differences, including 1–2 Å displacements of helices that contribute to the binding pocket. Although it is a misconception that crystallization chaperones can 'force' the transporter into a non-native, high-energy conformation (*Koide, 2009*), it is plausible that the monobody chaperones recognize a less-prevalent conformation, and kinetically trap the transporter in a minority state within the native conformational ensemble. Because these monobodies were not selected against EmrE, but against a different homologue from the SMR family, this is a possibility that should be considered. However, two lines of evidence disfavor the possibility that the monobody-bound state is aberrant. First, we showed that monobody binding has only a minor effect on transport function, and second, our model corresponds closely to the helix density in the EM dataset, which was obtained without exogenous binding proteins (*Ubarretxena-Belandia et al., 2003*). Although local perturbations at the monobody-binding interface of loops $1_A$ and $1_B$ cannot be ruled out, the position of loop $1_A$ is consistent with prior spectroscopic data, which predicted that in the major solution conformation, $F27_A$ packs against the B subunit with its sidechain oriented toward the substrate-binding site (*Dastvan et al., 2016*). Loop $1_B$ is located on the open side of the transporter and does not form any intra-transporter contacts. Therefore, even if monobody does stabilize a less-prevalent conformation of loop $1_B$, this would not change the major interpretations of the present structures.

## Comparison to the NMR model of EmrE S64V

An NMR-based model of the 'slow-exchanging' EmrE mutant S64V was recently published (*Shcherbakov et al., 2021*). S64V binds substrate with similar affinity as wildtype, but the rate of conformational exchange is about an order of magnitude slower (*Wu et al., 2019*). This model was computed based on chemical shift measurements and distance restraints between the protein backbone and the fluorinated substrate tetrafluorophenyl phosphonium (F-TPP$^+$). Although our present crystal structures agree with the NMR model in general aspects, such as the antiparallel topology, there are also notable differences in the global conformation, with an overall RMSD of 2.3 Å for the two models. Relative to other models of EmrE, including the computational models (*Ovchinnikov et al., 2018*; *Vermaas et al., 2018*), the EM α-helical model (*Ubarretxena-Belandia et al., 2003*), and the present crystal structures, in the NMR model the first lobe of the A subunit is shifted down in a direction perpendicular to the membrane with respect to the B subunit (*Figure 6A*). Note that chain A of the NMR structure is more structurally homologous to chain B of the crystal structure and vice versa. Our designation of chains A and B in the present crystal structure correspond to the A and B chains in previous literature, including SMR family homologue Gdx-Clo (*Kermani et al., 2020*), the low-resolution EmrE structures of EmrE (*Chen et al., 2007*; *Fleishman et al., 2006*), and theoretical EmrE models (*Ovchinnikov et al., 2018*; *Vermaas et al., 2018*) This difference in subunit packing is accompanied by subtle differences in the tilts of the helices (*Figure 6B*). In the NMR structure, helix $2_A$ and $2_B$ become more parallel, and the gap between them is narrowed, reducing membrane access to the binding site via the portal.

The difference in global conformation of the NMR and crystallography models is supported by a reorganization of the hydrogen bonding network in the substrate binding site (*Figure 6C*). The heart of this change is a rotameric switch by Y60: In the crystal structures, $Y60_B$ participates in a pair of cross-subunit interactions, within coordination distance and geometry of $E14_A$ and $W63_A$ in the opposite subunit. In the NMR model, the same $Y60_B$ sidechain is assigned a different rotamer, its hydroxyl moving 6 Å along helix 1, so that it is now coordinating $T18_A$, one helical turn away from $E14_A$. The interaction with $Y60_B$ has displaced $Y40_A$ from its interaction with $T18_A$. Helix $2_A$ slides in a direction perpendicular to the membrane so that $Y40_A$ now encroaches on the position of $F27_A$ at the tip of loop 1, which is packed between helices $2_A$ and $2_B$ in the crystal structure. In the NMR ensemble, the displaced loop one is flexible and adopts various conformations. The helix density observed in the low-resolution EM dataset corresponds closely to the present crystallography models (Real space correlation coefficient (RSCC) = 0.67), and is less consistent with the NMR model (RSCC = 0.51; *Figure 6—figure supplement 1*; *Ubarretxena-Belandia et al., 2003*).

The differences in conformation between the crystallography/EM datasets and the NMR model are unlikely to be due to membrane mimetic (which is shared for the EM and NMR datasets), the presence of monobodies (the EM data was collected without monobodies), or the S64V mutation used for NMR

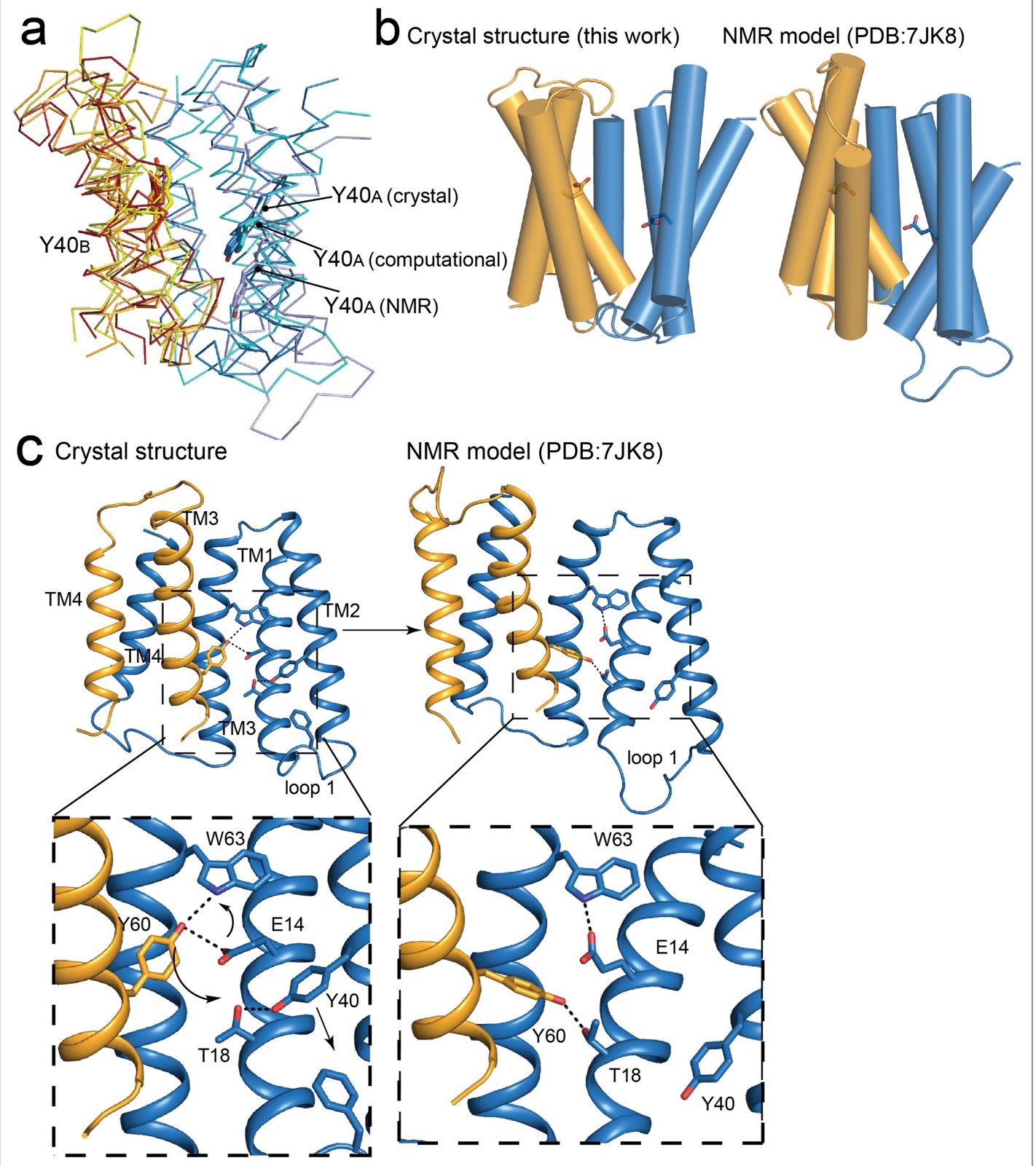

**Figure 6.** Comparisons of NMR and crystallography models of EmrE. (**A**) Overlay of crystallography (orange/blue; computational (yellow/cyan; *Vermaas et al., 2018*) and NMR (dark red/pale blue; *Shcherbakov et al., 2021*) models, aligned over the B subunit. Y40 sidechain sticks are show as landmarks. (**B**) Side-by-side comparisons of the crystallography and NMR models, with A subunit in blue and B subunit in orange. E14 sidechains shown as landmarks. (**C**) Comparison of Y60_B hydrogen bonding network in the crystal structure (left) and NMR structure (right). EmrE dimers are shown with TM

*Figure 6 continued on next page*

*Figure 6 continued*

1 and 2 of subunit B (orange) removed for clarity. Lower panels show zoomed in view. In each view, interactions within hydrogen bonding distance and geometry are shown as dashed lines. Arrows are shown to help visualize sidechain rearrangements between the two structures.

The online version of this article includes the following source data and figure supplement(s) for figure 6:

**Figure supplement 1.** Comparison of EmrE models with electron microscopy density.

**Figure supplement 2.** Comparison of experimental chemical shifts for EmrE (BMRB accession number 50411) with chemical shifts predicted from the crystallography model and NMR ensemble using LARMOR$^{C\alpha}$ (*Frank et al., 2015*).

**Figure supplement 2—source data 1.** Chemical shift predictions for NMR model and crystallography model and NMR ensemble using LARMOR$^{C\alpha}$.

studies (NMR experiments showed little change in backbone configuration for this mutant *Wu et al., 2019*). It is possible that the elevated temperature of the NMR experiments (45 ° C, compared to 20 ° C for crystallization) favor different states in a conformational ensemble. Previous EPR measurements may lend support to this possibility (*Dastvan et al., 2016*). Those experiments showed that at pH 8, with TPP$^+$ bound, EmrE adopts a major conformation consistent with our current crystallography model. But when substrate is removed and the pH dropped to 5.5, EmrE's conformational ensemble becomes more heterogeneous. The loops disengage and become more flexible, and a population emerges in which the two subunits have adopted a more-symmetric conformation. Perhaps the NMR experiments, which were performed at pH 5.5 (albeit with substrate) reflect that second conformation from the ensemble. Nevertheless, it is also worth noting that our crystallography model is not inconsistent with the backbone chemical shifts measured in bicelles based on structure-trained predictions of chemical shift (*Frank et al., 2015*; *Xie et al., 2020*).

## Comparison to prior functional studies of EmrE

EmrE has been studied in great breadth and depth. Full mutagenic scans coupled with growth assays (*Amadi et al., 2010*; *Gutman et al., 2003*; *Mordoch et al., 1999*; *Wu et al., 2019*), functional assays with reconstituted transporter (reviewed in *Schuldiner, 2009*), and EPR and NMR spectroscopy experiments *Amadi et al., 2010*; *Banigan et al., 2015*; *Dastvan et al., 2016*; *Leninger et al., 2019*; *Thomas et al., 2018* have all revealed detailed information about the positions that contribute to substrate binding and conformational change, even as the structural details were lacking. Our structure corroborates many of the specific predictions regarding sidechains that contribute to the binding pocket, including the importance of W63 for aromatic packing with the substrate (*Elbaz et al., 2005*) and the cross-subunit engagement of Y60 (*Vermaas et al., 2018*). Positions that are sensitive to mutation, including E14, T18, Y40, and L47 all line the binding pocket in our structures (*Mordoch et al., 1999*; *Rotem et al., 2006*; *Saleh et al., 2018*; *Wu et al., 2019*). Our structure also confirms other architectural features proposed from spectroscopic studies, including the deflection of loop two sidechain F27$_A$ toward the substrate bound in the binding pocket and the positioning of the portal-lining Y40 and F44 sidechains as an access point from the membrane to the substrate-binding site (*Dastvan et al., 2016*). Our results also provide some insight into the observation that a single L51I or I62L mutation in one subunit of the EmrE dimer prevents conformational exchange (*Leninger et al., 2019*). Both residues are located on transmembrane helices and are buried at protein interfaces in one monomer and accessible in the other (L51 to the aqueous binding pocket and I62 to the membrane). For Gdx-Clo, we previously posited that differential packing of the two monomers in the N-terminal half of helix three contributes to structural frustration and the resulting conformational exchange (*Kermani et al., 2020*). In EmrE, I62 is located in this same crucial region, and its mutation in only one monomer presumably disturbs the well-matched competition that occurs in the homodimer.

In addition to a substrate-free, pH 5.2 structure, we solved structures of EmrE with methyl viologen, harmane, Me-TPP$^+$, TPP$^+$, and benzyltrimethylammonium at pH values between 6.3 and 7.5. Experiments with EmrE in bicelles have suggested that a proton can bind simultaneously with TPP$^+$ with a pK$_a$ of 6.8 (*Robinson et al., 2017*). In the NMR model, under conditions that favor simultaneous substrate and proton binding, F-TPP$^+$ is positioned higher in the binding pocket, 2 Å closer to E14$_B$ than protonated E14$_A$ (*Shcherbakov et al., 2021*). In contrast, in our TPP$^+$-bound structure, which was obtained at a pH of 7.25, TPP$^+$ is situated lower in the binding pocket and within 0.5 Å of the midpoint between the glutamates. It is thus probable that this crystal structure represents the doubly-deprotonated, substrate-bound state. It is also likely that both glutamates are deprotonated in the

methyl viologen-bound structure, since this substrate bears a + 2 charge, making glutamate protonation more electrostatically unfavorable than in the presence of a monovalent substrate.

Protonation of the central glutamates has not been evaluated in the presence of monovalent substrates other than $TPP^+$, and the E14 $pK_a$ values are likely to vary according to factors such as binding pocket solvation or charge delocalization on the substrate. For the Me-$TPP^+$, harmane, and benzyltrimethylammonium-bound structures (pH 6.5, 7.1, and 7.25, respectively), the contribution of a substrate+ proton-bound population cannot be ruled out. However, the positioning of each of these substrates centered close to the midpoint between the E14 carboxylate groups, similar to $TPP^+$, implies that in the major component of the population, both glutamates bear a negative charge.

## Sidechain movements accommodate diverse substrates

In addition to substantiating prior EmrE experiments, our structures also provide new molecular insights into the binding of structurally diverse substrates by EmrE. Methyl viologen, harmane, Me-$TPP^+$, $TPP^+$, and benzyltrimethylammonium have considerable structural differences, but are all accommodated in the EmrE binding site with only sidechain rearrangements. The closely related, but substantially more selective SMR family member, Gdx-Clo, provides a useful point of comparison to understand why EmrE can interact with this chemically diverse range of compounds. In Gdx-Clo, the substrate-binding glutamate sidechains are constrained by a polarized stack of hydrogen bond donors and acceptors that also includes $W16_{Clo}$, $S42_{Clo}$, and $W62_{Clo}$. This hydrogen bonded network would be disrupted by the rotamerization of either $E13_{Clo}$ or $W62_{Clo}$. We show that in Gdx-Clo, mutations to sidechains that contribute to the hydrogen bond stack seriously impair transport activity.

In contrast, in EmrE, the corresponding residues $E14_{EmrE}$ and $W63_{EmrE}$ are not constrained by such a stack of H-bond donors and acceptors. The current structures and SSM electrophysiology experiments both suggest that, in contrast to Gdx-Clo, a rigid H-bond network is not essential for substrate transport by EmrE, which remains functional when hydrogen bond capacity is eliminated at $S43_{EmrE}$ or $W63_{EmrE}$. Without the stricter geometric constraints imposed by a polarized stack of sidechain hydrogen bond partners, both $E14_{EmrE}$ and $W63_{EmrE}$ have more flexibility to adopt different rotamers. Like a pair of calipers, the $E14_{EmrE}$ sidechains can move farther apart to accommodate large substrates such as quaternary ammoniums, or closer together for flat, aromatic substrates or substrates with small headgroups, like harmane and methyl viologen or singly substituted guanidinyl compounds. Similarly, $W63_{EmrE}$ has the space and flexibility to rotamerize, which can expand or narrow the binding pocket or allow $W63_{EmrE}$ to pack against the aromatic groups of bound substrates. These structural observations are in agreement with numerous prior studies that have demonstrated an important role for $W63_{EmrE}$ in transport of polyaromatic substrates (*Amadi et al., 2010*; *Elbaz et al., 2005*; *Saleh et al., 2018*; *Wu et al., 2019*). We note that although $W63_{A, EmrE}$ does change position in order to conform to different substrates, we did not always observe optimal pi stacking geometry between the substrate and the protein's aromatic residues. Instead, substrate positioning appeared to optimize electrostatic interactions first, with all substrates situated directly between $E14_{A, EmrE}$ and $E14_{B, EmrE}$.

Likewise, many EmrE substrates lack the capacity to donate strong hydrogen bonds, reducing the geometric constraints for protein-substrate interactions. Prior MD simulations suggested a dynamic interaction between $TPP^+$ and the EmrE-binding pocket (*Vermaas et al., 2018*), and we expect that many compounds transported by EmrE have some mobility within the binding pocket. In the present structural experiments, we observe this explicitly for methyl viologen, which we identified in different but overlapping positions in the two transporters in the asymmetric unit.

While our experiments indicate that altering sidechain configuration is important to accommodate diverse substrates, backbone conformational changes do not need to be invoked to explain polyspecificity. Indeed, we do not see perturbations in EmrE's main chain structure in the six different EmrE crystal structures resolved here. In addition, the general correspondence of the structures of EmrE and Gdx-Clo indicates that same tertiary architecture can also accommodate substrates with guanidinyl headgroups and/or alkyl tails. These observations also concur with observations from cryo-EM, which showed only minor differences in helix orientation and packing for the apo and $TPP^+$-bound structures (*Tate et al., 2003*). Thus, the crystallized conformation can accommodate substrates from major classes, including quaternary ammoniums, quaternary phosphoniums, planar polyaromatics, and substituted guanidiniums without substantial backbone rearrangement.

## Binding of benzalkonium⁺ and other substrates with alkyl chains

Because benzalkonium is especially relevant as a common household and hospital antiseptic to which the Qac proteins provide resistance, we sought to visualize how this quaternary ammonium compound might interact with EmrE. Although we were unable to generate diffracting crystals of EmrE$_3$ in the presence of substrates with long alkyl tails, our current structure of EmrE$_3$ with benzyltrimethylammonium bound (a chemical homologue of benzalkonium with a methyl group in place of the alkyl chain), combined with our previous Gdx-Clo structure, provides a strong indication of how benzalkonium or other detergent-like substrates might bind.

In Gdx-Clo, octylGdm⁺ binds such that its alkyl tail extends out of the aqueous binding pocket and into the membrane. In order to accommodate the alkyl tail, hydrophobic sidechains lining Gdx-Clo's TM2 portal, including M39$_{Clo}$ and F43$_{Clo}$, adopted alternative rotamers (*Kermani et al., 2020*). Although all the substrates in the present EmrE$_3$ structures were contained within the aqueous pocket, we similarly observe rotameric rearrangements of the TM2 sidechains in different structures, including Y40$_{EmrE}$ and F44$_{EmrE}$ (equivalent to Gdx-Clo's M39$_{Clo}$ and F43$_{Clo}$) in the harmane and methyl viologen structures. These observations suggest that, as in Gdx-Clo, in EmrE the sidechain packing at the TM2 interface is malleable, and that movements of these residues may remodel the TM2 portal to permit binding of substrates with detergent-like alkyl chains.

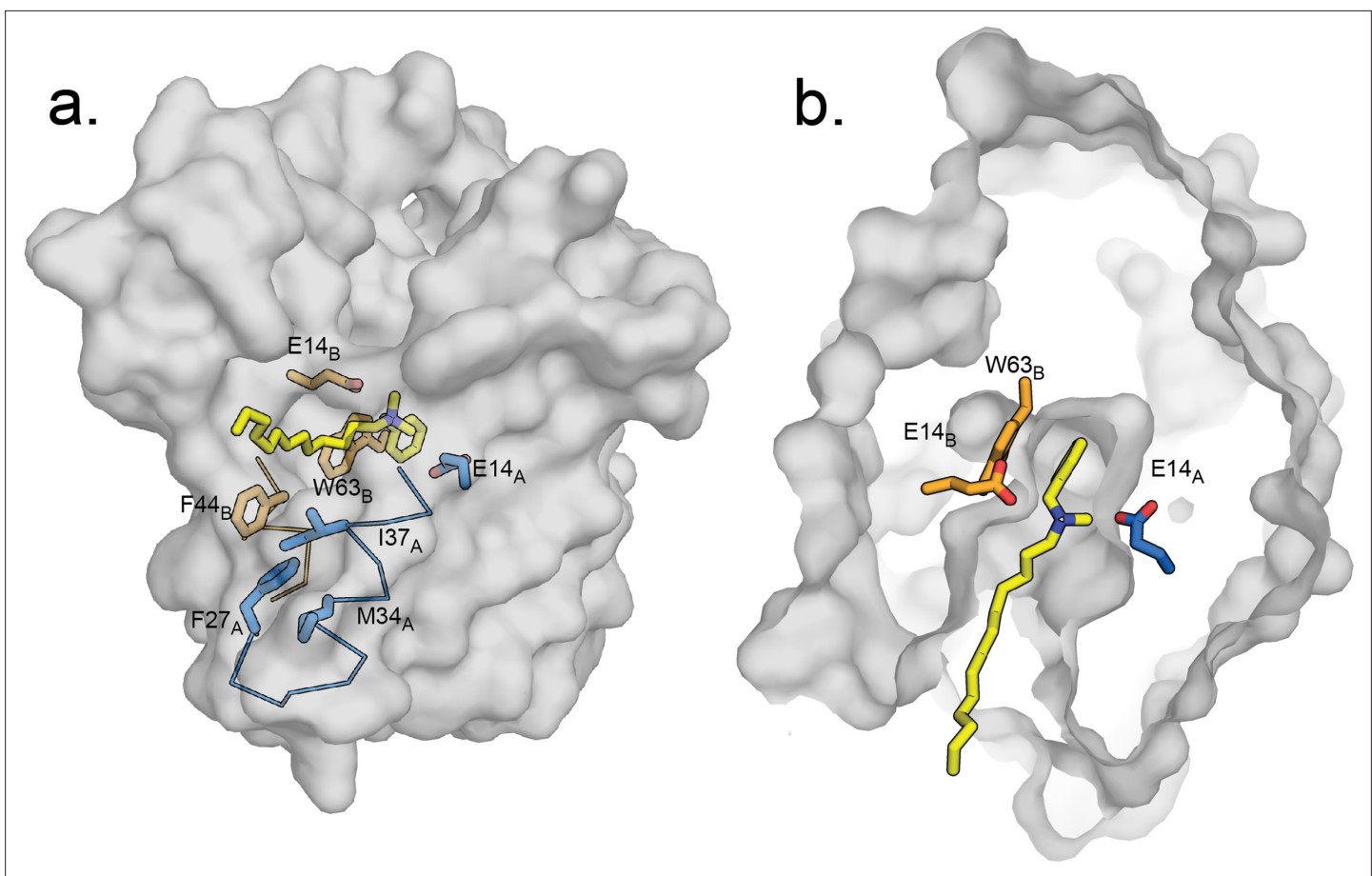

**Figure 7.** Hypothetical model of benzalkonium binding to EmrE. (**A**) Benzalkonium is shown in yellow stick representation. Sidechains from the A and B subunits are colored as before. The mainchain for helices lining the TM2 portal is shown in ribbon format, with the portal-lining sidechains shown as sticks. (**B**) Top-down view of binding site with benzalkonium. EmrE is sliced at the midpoint of the membrane. Comparisons of this model to the experimental models of EmrE in complex with benzyltrimethylammonium (PDB:7T00) and Gdx-Clo in complex with octylguanidinium (PDB:6WK9) are shown in *Figure 7—figure supplement 1*.

The online version of this article includes the following figure supplement(s) for figure 7:

**Figure supplement 1.** Top down structures of EmrE in complex with benzyltrimethylammonium (PDB:7T00; model for benzalkonium headgroup binding) and Gdx-Clo in complex with octylguanidinium (PDB:6WK9; model for alkyl tail positioning).

Indeed, when the quaternary ammonium headgroup of benzalkonium is superposed onto the experimentally determined position of benzyltrimethylammonium in the $EmrE_3$ binding pocket, the alkyl tail of benzalkonium extends towards the portal defined by the TM2 helices. Although the extended alkyl chain would clash with $F44_{B, EmrE}$, positioning this sidechain in the 'down' rotamer (analogous to that adopted by $F43_{B, Clo}$ in Gdx-Clo) alleviates all clashes between the substrate and protein and provides unobstructed access for the alkyl tail to the membrane interior. *Figure 7* shows a proposed model of benzalkonium binding to EmrE prepared by aligning its headgroup with benzyltrimethylammonium followed by energy minimization of the complex using MMTK (*Hinsen, 2000*).

Thus, we propose that sidechain rearrangements along the membrane portal also contribute to substrate polyspecificity by allowing hydrophobic substituents to extend out of the substrate-binding site and access the membrane interior. Similarly, we imagine that dipartite drugs transported by EmrE, such as propidium (a planar polyaromatic group linked to a tetraethyl ammonium) and dequalinium (two aromatic groups with a 10-carbon linker) may also utilize the portal for transport, with the protein-mediated transport of one moiety dragging its tethered lipophilic partner across the membrane.

## Conclusions

In summary, we have developed a multipurpose crystallization chaperone for SMR proteins and used this tool to resolve the first sidechain-resolution crystal structures of the bacterial SMR transporter, EmrE. In order to establish the structural basis of substrate polyspecificity, we resolved structures with five different substrates bound, including quaternary phosphoniums, planar aromatics, and a quaternary ammonium compound. We propose that, compared with more selective representatives of the SMR family, a relatively sparse hydrogen bond network among binding site residues in EmrE permits sidechain flexibility to conform to structurally diverse substrates.

# Materials and methods

**Key resources table**

| Reagent type (species) or resource | Designation | Source or reference | Identifiers | Additional information |
|---|---|---|---|---|
| Gene (*Escherichia coli*) | $EmrE_3$ | Uniprot | P23895 | Bears mutations E25N, W31I, V34M to bind monobody (this paper – see *Figure 1*) |
| Gene (*Clostridiales* bacterium oral taxon 876) | Gdx-Clo | GenBank | ERI95081.1 | PMID:33247110 |
| Recombinant DNA reagent | $EmrE_3$ in pET15b (plasmid) | This publication | | Expression vector for $EmrE_3$. Available upon request. |
| Recombinant DNA reagent | Gdx-Clo in pET21c (plasmid) | PMID:33247110 | | Expression vector for Gdx-Clo. Available upon request. |
| Chemical compound, drug | *E. coli* polar lipids | Avanti, Alabaster, AL | #100600 C | |
| Chemical compound, drug | n-decyl-β -D-maltopyranoside | Anatrace, Maumee, OH | D322 | |
| Recombinant DNA reagent | Monobody L10 in pHBT1 (plasmid) | PMID:33247110 | | Expression vector for monobody L10. PMID:33247110. Addgene ID: 183,406 |

## Bioinformatics and sequence analysis

Multiple sequence alignment was performed using MUSCLE (*Edgar, 2004*). ConSurf was used for sequence conservation analysis (*Ashkenazy et al., 2016*; *Berezin et al., 2004*). For this analysis, SMR sequences from GEBA bacterial reference genomes (*Mukherjee et al., 2017*) that were identified as probable homodimers based on genetic context (those encoded by a single gene in an operon) were further sorted into either Qac or Gdx subclasses using profile Hidden Markov Models built from the corresponding sequence clusters of the functionally annotated sequence similarity network described in *Kermani et al., 2020*. Representative sequences were selected for the alignments in *Figures 1 and 4* because (1) proteins have been characterized in transport or resistance assays and (2) sequences are distributed among different major clades of the phylogenetic tree (*Kermani et al., 2018*).

## Protein purification and crystallization

L10 monobody was purified from inclusion bodies exactly as described in detail previously (**Kermani et al., 2020**). pET15b plasmids bearing the EmrE$_3$ coding sequence with an N-terminal hexahistidine tag and a thrombin cut site were transformed into *E. coli* C41 and grown overnight (15–18 hr) in Studier's autoinduction media at 37 °C. Pellets were resuspended in breaking buffer (50 mM Tris-Cl pH 8.0, 100 mM NaCl, 10 mM tris(2-carboxyethyl)phosphine (TCEP)) with 400 µg DNase, 2 mM MgCl$_2$, 1 mM PMSF, 1 mg/mL lysozyme, 25 µg pepstatin, and 500 µg leupeptin. Resuspended pellets were lysed by sonication and extracted with 2% n-Decyl-β-D-Maltopyranoside (DM) (Anatrace) for 2 hr at room temperature. Extract was clarified by centrifugation (16,000 rpm, 4 °C, 45 min), and loaded onto TALON cobalt resin equilibrated with wash buffer (20 mM tris-Cl pH 8.0, 100 mM NaCl, 5 mM DM) supplemented with 5 mM TCEP. Column was washed with wash buffer, and wash buffer supplemented with 10 mM imidazole before elution of EmrE$_3$ with wash buffer supplemented with 400 mM imidazole. After exchange into wash buffer using PD-10 desalting columns (GE Healthcare) His tags were cleaved with thrombin (1 U/mg EmrE$_3$) overnight at room temperature (21 °C) prior to a final size exclusion purification step using a Superdex 200 column equilibrated with 10 mM 2-[4-(2-hydroxyethyl)piperazin-1-yl]ethanesulfonic acid (HEPES) pH 7.5, 100 mM NaCl, 4 mM DM.

For functional measurements, protein was reconstituted by dialysis as previously described (**Kermani et al., 2020**). For SSM electrophysiology experiments, proteoliposomes were prepared with 20 mg EPL per ml, and a 1:20 protein:lipid mass ratio. Proteoliposomes were aliquoted and stored at –80 ° C until use. For crystallography of EmrE$_3$, monobody L10 and EmrE$_3$ were each concentrated to 10 mg/mL, and the L10 protein solution was supplemented with 4 mM DM. EmrE$_3$ and L10 were combined in a 2.1:1 molar ratio and supplemented with lauryldimethylamine oxide (LDAO, final concentration of 6.6 mM). The protein solution was mixed with an equal volume of crystallization solution (0.3 µL in 96-well plates). Crystals formed after approximately 4 weeks, and were frozen in liquid nitrogen before data collection. For crystallization with substrate, the EmrE$_3$/monobody/LDAO solution was prepared as before, and substrate was added from a stock solution immediately before setting crystal trays (final concentrations of 1 mM for methyl viologen, 500 µM for harmane, 300 µM for benzyltrimethylammonium, 100 µM for TPP$^+$, or 300 µM for MeTPP$^+$). The low pH EmrE$_3$ crystals grew in 200 mM NaCl, 100 mM sodium cacodylate, pH 5.2, 34% PEG 600. The substrate-bound EmrE$_3$ crystals grew in 100 mM LiNO$_3$ or 100 mM NH$_4$SO$_4$, 100 mM ADA, pH 6.5 or 100 mM HEPES, pH 7.1–7.3, and 30–35% PEG 600. Gdx-Clo protein and crystals were prepared exactly as described previously (**Kermani et al., 2020**). Crystals grew in 100 mM calcium acetate, 100 mM sodium acetate, pH 5.0, 40% PEG600.

## Structure determination and analysis

Crystallography data was collected at the Life Sciences Collaborative Access Team beamline 21-ID-D at the Advanced Photon Source, Argonne National Laboratory. Diffraction data were processed and scaled using Mosflm 7.3 (**Battye et al., 2011**) or DIALS (**Winter et al., 2018**). Crystals diffracted anisotropically, and electron density maps were improved by anisotropic truncation of the unmerged data using the Staraniso webserver (**Tickle et al., 2018**) with a cutoff level of 1.2–1.8 for the local $I/\sigma< I >$ . For the low pH EmrE$_3$ dataset, phases were determined using molecular replacement with Phaser (**McCoy et al., 2007**), using the first three helices of Gdx-Clo and the L10 monobody structures (PDB:6WK8) as search models. Loop 3, helix 4, and the C-terminal loop were built into the experimental electron density using Coot (**Emsley et al., 2010**), with iterative rounds of refinement in Phenix (**Liebschner et al., 2019**) and Refmac (**Murshudov et al., 2011**). For the low pH Gdx-Clo structure, Gdx-Clo and the L10 monobody structures (PDB:6WK8) were used as molecular replacement search models. Models were validated using Molprobity (**Williams et al., 2018**) and by preparing composite omit maps in Phenix, omitting 5% of the model at a time (**Terwilliger et al., 2008**). The substrate-bound structures were phased using molecular replacement with monobody L10 and the A and B subunits of the initial EmrE$_3$ model as the search models. Proteins typically crystallized in C121, although the methyl viologen-bound EmrE$_3$ structure and the low pH Gdx-Clo crystallized in P1. For both, the unit cell contained two pseudosymmetric copies of the transporter-monobody complex. The angle of the bend in TM3 was analyzed using Kink Finder (**Wilman et al., 2014**).

## Microscale thermophoresis

Monobody L10 was labeled at a unique, introduced cysteine, A13C, with fluorescein maleimide. Binding to $EmrE_3$ was measured using microscale thermophoresis (Nanotemper, Munich, Germany). For these experiments, labeled monobody was held constant at 2 µM, and the concentration of $EmrE_3$ was varied from 30 nM to 100 µM. Buffer contained 100 mM NaCl, 10 mM HEPES, pH 7, 4 mM DM, and 50 µg/mL bovine serum albumin. Samples were incubated at least 30 min prior to measurement of binding interactions. Experiments were performed using three independent sample preparations and fit to a one site binding equilibrium with total L10 as the experimental variable:

$$MST\left([EmrE]\right) = MST_0 + \frac{(MST_f - MST_0)}{2}\left(1 + \frac{[EmrE]}{[L10]} + \frac{K_D}{[L10]}\right)\left[1 - \sqrt{1 - \frac{4\frac{[EmrE]}{[L10]}}{\left(1 + \frac{[EmrE]}{[L10]} + \frac{K_D}{[L10]}\right)^2}}\right] \quad (1)$$

where MST([EmrE]) is the MST signal as a function of total EmrE added to a fixed concentration of labelled L10 monobody, and $MST_0$ and $MST_f$ are the arbitrary initial and final MST fluorescence signals.

## SSM electrophysiology

SSM electrophysiology was conducted using a SURFE²R N1 instrument (Nanion Technologies, Munich, Germany) according to published protocols (*Bazzone and Barthmes, 2020*; *Bazzone et al., 2017*). The sensor was alkylated and painted with lipid solution (7.5 µg/µl 1,2-diphytanoyl-sn-glycero-3-phos phocholine in n-decane), followed immediately by addition of recording buffer (100 mM KCl, 100 mM $KPO_4$, pH 7.5). For measurements in the presence of monobody, buffers also contained 50 µg bovine serum albumin/mL. Proteoliposomes were applied to the sensor surface and centrifuged at 2500 x g for 30 min. Before experiments, sensors were checked for conductance and capacitance using SURFE²R software protocols. Sensors for which capacitance and conductance measurements were outside an acceptable range (10–40 nF capacitance, 1–5 nS conductance) were not used for experiments. Sensors were periodically rechecked for quality during the course of an experiment. When multiple measurements were performed on a single sensor, currents elicited by a reference compound were measured at the outset of the experiment and again after collecting data on test compounds. If currents differed by more than 10% between the first and last perfusions, this indicated that the proteoliposomes associated with the sensor had not remained stable over the course of the experiment, and data collected in this series was discarded. Between measurements, sensors were perfused with substrate-free solution for 2 s; observation of capacitive currents with opposite polarity indicated substrate efflux from the proteoliposomes and a return to the resting condition.

## NMR chemical shift prediction

The chemical shifts of the $C_\alpha$ atoms of the NMR ensemble and the unliganded crystallography model were predicted using LARMOR$^{C\alpha}$ (*Frank et al., 2015*) as implemented with PyShifts (*Xie et al., 2020*).

# Acknowledgements

We thank the Stockbridge lab for comments on the project and manuscript, and we are grateful to Aaron Frank (University of Michigan) for helpful conversations about chemical shift-based comparisons of the structures. Funding: This work was supported by NSF CAREER award 1845012 to RBS and R01 CA194864 to SK. This research used resources of the Advanced Photon Source, a U.S. Department of Energy (DOE) Office of Science User Facility operated for the DOE Office of Science by Argonne National Laboratory under Contract No. DE-AC02-06CH11357. Use of the LS-CAT Sector 21 was supported by the Michigan Economic Development Corporation and the Michigan Technology Tri-Corridor (Grant 085P1000817). RBS is a Burroughs Wellcome Fund Investigator in the Pathogenesis of Infectious Disease.

# Additional information

#### Competing interests

Akiko Koide: is listed as inventor for patents (US9512199 B2 and related patents and applications) covering aspects of the monobody technology filed by the University of Chicago and Novartis. Shohei Koide: is listed as inventor for patents (US9512199 B2 and related patents and applications) covering aspects of the monobody technology filed by the University of Chicago and Novartis. Is a scientific advisory board member and holds equity in and receives consulting fees from Black Diamond Therapeutics; receives research funding from Puretech Health and Argenx BVBA. Randy B Stockbridge: Reviewing editor, *eLife*. The other authors declare that no competing interests exist.

## Funding

| Funder | Grant reference number | Author |
| --- | --- | --- |
| National Institutes of Health | CA194864 | Shohei Koide |
| National Science Foundation | CAREER 1845012 | Randy B Stockbridge |
| Burroughs Wellcome Fund | | Randy B Stockbridge |

The funders had no role in study design, data collection and interpretation, or the decision to submit the work for publication.

## Author contributions

Ali A Kermani, Conceptualization, Formal analysis, Investigation, Methodology, Writing – review and editing; Olive E Burata, Conceptualization, Formal analysis, Investigation, Methodology, Visualization, Writing – review and editing; B Ben Koff, Investigation; Akiko Koide, Investigation, Methodology; Shohei Koide, Funding acquisition, Methodology, Writing – review and editing; Randy B Stockbridge, Conceptualization, Formal analysis, Funding acquisition, Methodology, Project administration, Supervision, Visualization, Writing - original draft

## Author ORCIDs

Olive E Burata http://orcid.org/0000-0002-8450-8930
B Ben Koff http://orcid.org/0000-0003-3276-143X
Randy B Stockbridge http://orcid.org/0000-0001-8848-3032

## Decision letter and Author response

Decision letter https://doi.org/10.7554/eLife.76766.sa1
Author response https://doi.org/10.7554/eLife.76766.sa2

# Additional files

## Supplementary files
- Transparent reporting form

## Data availability

Atomic coordinates for the crystal structures have been deposited in the Protein Data Bank under accession numbers 7MH6 (EmrE3/L10), 7MGX (EmrE3/L10/methyl viologen), 7SVX (EmrE3/L10/ harmane), 7SSU (EmrE3/L10/MeTPP+), 7SV9 (EmrE3/L10/TPP+), 7T00 (EmrE3/L10/benzyltrimethylammonium) and 7SZT (Gdx-Clo/L10). All other data generated or analyzed during this study are included in the manuscript and supporting file; source data files have been provided for Figures 1 and 5.

The following datasets were generated:

| Author(s) | Year | Dataset title | Dataset URL | Database and Identifier |
|---|---|---|---|---|
| Kermani AA, Stockbridge RB | 2022 | Structure of EmrE-D3 mutant in complex with monobody L10 in low pH (protonated state) | https://www.rcsb.org/structure/7MH6 | RCSB Protein Data Bank, 7MH6 |
| Kermani AA, Stockbridge RB | 2022 | Structure of EmrE-D3 mutant in complex with monobody L10 and methyl viologen | https://www.rcsb.org/structure/7MGX | RCSB Protein Data Bank, 7MGX |
| Kermani AA, Stockbridge RB | 2022 | Structure of EmrE-D3 mutant in complex with monobody L10 and harmane | https://www.rcsb.org/structure/7SVX | RCSB Protein Data Bank, 7SVX |
| Kermani AA, Stockbridge RB | 2022 | Structure of EmrE-D3 mutant in complex with monobody L10 and methyltriphenylphosphonium | https://www.rcsb.org/structure/7SSU | RCSB Protein Data Bank, 7SSU |
| Kermani AA, Stockbridge RB | 2022 | Structure of EmrE-D3 mutant in complex with monobody L10 and TPP | https://www.rcsb.org/structure/7SV9 | RCSB Protein Data Bank, 7SV9 |
| Kermani AA, Stockbridge RB | 2022 | Structure of EmrE-D3 mutant in complex with monobody L10 and benzyltrimethylammonium | https://www.rcsb.org/structure/7T00 | RCSB Protein Data Bank, 7T00 |
| Kermani AA, Stockbridge RB, Burata OE | 2022 | Crystal structure of Gdx-Clo from Small Multidrug Resistance family of transporters in low pH (protonated state) | https://www.rcsb.org/structure/7SZT | RCSB Protein Data Bank, 7SZT |

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
