## [Editor Report]

*E. coli* EmrE and other members of the SMR family of transporters utilize the proton motive force to pump out a broad spectrum of antibiotics, thereby contributing to multi-drug resistance. Here, a new multipurpose crystallization chaperone is used to determine the structure of EmrE in apo form and in complex with various substrates. The strength of the manuscript is in the description of six new structures of EmrE at a resolution sufficient for building an atomic model and understanding how the antimicrobial agents bind, allowing robust conclusions to be drawn regarding the molecular details of binding of the antimicrobial agents. The report will be of interest to both those studying antibiotic resistance and those studying transporters.

---

## [Decision Letter]

**Decision letter after peer review:**

Thank you for submitting your article "Crystal structures of bacterial Small Multidrug Resistance transporter EmrE in complex with structurally diverse substrates" for consideration by *eLife*. Your article has been reviewed by 3 peer reviewers, including Nir Ben-Tal as Reviewing Editor and Reviewer #1, and the evaluation has been overseen by Volker Dötsch as the Senior Editor. The following individuals involved in review of your submission have agreed to reveal their identity: Christopher G Tate (Reviewer #2); Katherine A Henzler-Wildman (Reviewer #3).

Essential revisions:

We would appreciate more discussion of two points:

1. The L10 monobody has provided a valuable tool to facilitate structure determination of SMR family members, and we agree the EmrE mutations are likely not functionally significant. However, we would appreciate more discussion of the potential impact of monobody interaction itself on the EmrE structure, particularly in the loop regions. This was discussed in more detail in the prior Gdx-Clo structure paper from the Stockbridge group, but revisiting this is important here because of the role the loops play in closing off the transporter on one side of the membrane. Also, can this structure provide insight into how the L51I (and I62L) mutation near the end of TM2 (Lehninger *eLife* 2019;8:e48909) preferentially stabilizes EmrE in an open-to-one side conformation?

2. What is the exact pH at which each of the substrate-bound crystal structures were determined? The methods state that the substrate-bound structures (drug-like-substrate, proton is also a substrate) were crystallized at pH 6.5 or pH between 7.1-7.3 depending on the buffer. But we do not see the exact pH listed in the tables for each of the different crystal forms or discussed in the text. This information is important because of the proton-coupled transport mechanism.

The results state "To understand how different substrates interact with EmrE, we screened a variety of transported compounds in crystallization trials at pH values {greater than or equal to} 6.5, where the E14 sidechains are expected to be deprotonated, favoring binding of the positively charged substrates." This is incorrect. For EmrE bound to tetraphenylphosphonium, the pKa of the E14 residue that remains protonatable 6.8 {plus minus} 0.1, thus crystallization conditions near neutral pH may result in a mixture of drug-substrate-bound and drug-substrate-plus-proton bound transporter. The pH values should be listed more explicitly for each structure and it would be helpful to discuss the implication of the pH for mechanistic interpretation of the structures.

*Reviewer #1:*

The discovery of the multi-drug resistant transporter EmrE years ago has raised hopes that, being significantly smaller than other multi-drug transporters, it can be used as a simple model. As it turned out, however, it has been rather difficult to figure out its transport mechanism, and until we (Fleishman et al., 2006) modeled its structure based on cryo-EM data, even its membrane topology (being dual) was unclear. Here, Stockbridge and colleagues finally managed to fulfill the early hopes. Taking advantage of a new in house developed multipurpose crystallization chaperone, they managed to determine the high-resolution structure of apo-EmrE as well as co-complexes with various substrates. Remarkably, they noticed that EmrE's backbone remain more or less unaltered and that the ability to accommodate chemically diverse substrates is mostly due to sidechains that change rotameric states. They also determined the structure of a homologue with less broad substrates spectrum and saw that compared to this reference transporter, whose binding site residues are held fixed by hydrogen bonds, EmrE's equivalent residues are engaged in much less hydrogen bonds and much more flexible.

Main strengths here are characterization of so many high-resolution structures of EmrE, and the addition of functional assays showing that the constructs used are functional and physiologically relevant. Additional strength is the clever use of these structures and the structures of the homologue to decipher the transport mechanism is detail. Finally, another strength is that the multipurpose crystallization chaperone may be useful for other members of the family, and the overall approach for other multi-drug transporters.*Reviewer #2:*

The strength of the manuscript is in the description of six new structures of EmrE at a resolution sufficient for building an atomic model and understanding how the antimicrobial agents bind. What is notable is that the X-ray structures fit extremely well to a low resolution density map of EmrE determined by electron cryo-microscopy of the transporter embedded in a lipid bilayer, suggesting that the structure is a good representation of a physiologically relevant state. This is in contrast to a recently determined NMR structure of EmrE that does not fit the density so well. The quality of the X-ray structures allows the positions of amino acid side chains to be determined, thus allowing robust conclusions to be drawn regarding the molecular details of binding of the antimicrobial agents. The results are supported by extensive previously published work on EmrE performed by site-directed mutagenesis and activity assays. The manuscript is very clearly written and presented. I do not find any significant weaknesses in the manuscript.

*Reviewer #3:*

This work builds on the Stockbridge previous efforts to determine the structure of Gdx, an SMR homolog that has a much narrower substrate profile and functions as a toxic metabolite exporter (specifically guanidinium), rather than promiscuous toxin exporter like EmrE. Strength of the work is the large number of structures for EmrE, the SMR for which the function and transport mechanism has been most extensively studied. Structural data is important for understanding the molecular basis for promiscuous multidrug recognition and transport.

Given the resolution is in the 3-4Å range for the substrate bound structures, it is important to examine the density of not just the substrate, but also key sidechains in the active site when assessing how EmrE binds different substrates. I appreciate that this density is shown for the higher resolution TPP+ and methyl viologen bound structures to really demonstrate the strength of the conclusions that E14 and W63 change position to accommodate planar vs tetrahedral substrates.

Another strength is the comparison of the low pH EmrE and Gdx structures illustrating the difference in hydrogen bonding network within the central pore region, and the insights this provides into the difference in side chain orientations and promiscuity of these two homologous transporters. These insights into the substrate recognition mechanisms are important for understanding the divergence of the two SMR transporter subfamilies and may be helpful for understanding how proteins achieve promiscuous vs specific binding more broadly.

Two areas require further detail and discussion to be added to the manuscript:

1. While I agree that the L10 monobody has provided a valuable tool to facilitate structure determination of SMR family members, and the individual point mutations made to enable monobody binding are unlikely to cause significant functional or structural defects in EmrE, I would appreciate more discussion of the potential impact of the monobody on EmrE structure? The comparison with prior EM data is consistent with the monobody not having a significant impact on the overall arrangement of transmembrane helices, but I am more concerned with potential impact on the flexible loop regions. This was discussed in more detail in the prior Gdx-Clo structure paper from the Stockbridge group, but revisiting this is important here. This is particularly relevant because of the role the loops play in closing off the transporter on one side of the membrane. Figures highlighting the density of the loop regions on both the closed and open face of the transporter with the monobody interactions highlighted would be helpful in assessing this. In addition, it would be worthwhile to consider whether this structure can provide insight into how the L51I (and I62L) mutation near the end of TM2 (Lehninger *eLife* 2019;8:e48909) preferentially stabilizes EmrE in an open-to-one side conformation.

2. What is the exact pH at which each of the substrate-bound crystal structures were determined? The pH of the apo crystal structure is listed in the tables and discussed in the text of the manuscript, highlighting that this is a low-pH structure and thus represents the proton-bound state of the transporter (not truly apo). The methods state that the (drug-like, proton is also a substrate) substrate-bound structures were crystallized at pH 6.5 or pH between 7.1-7.3 depending on the buffer. But I do not see the exact pH listed in the tables for each of the different crystal forms or discussed in the text. This information is important because EmrE is a proton-coupled transporter.

The results state "To understand how different substrates interact with EmrE, we screened a variety of transported compounds in crystallization trials at pH values {greater than or equal to} 6.5, where the E14 sidechains are expected to be deprotonated, favoring binding of the positively charged substrates." This is not correct. The pKa values of E14 in drug-free EmrE are 7.0 and 8.2 at 25 {degree sign}C (Morrison, J Gen Phys 2015, 146:445). EmrE is able to bind drug-like substrate and proton simultaneously, although only one of the E14 remains protonatable (Robinson, PNAS 2017, 114:E10083). For EmrE bound to tetraphenylphosphonium, the pKa of the E14 residue that remains protonatable 6.8 {plus minus} 0.1, thus crystallization conditions near neutral pH may result in a mixture of drug-substrate-bound and drug-substrate-plus-proton bound transporter.

---

## [Author Response]

Essential revisions:We would appreciate more discussion of two points:1. The L10 monobody has provided a valuable tool to facilitate structure determination of SMR family members, and we agree the EmrE mutations are likely not functionally significant. However, we would appreciate more discussion of the potential impact of monobody interaction itself on the EmrE structure, particularly in the loop regions.

We have added a comment to this paragraph commenting on the loops explicitly:

“However, two lines of evidence disfavor the possibility that the monobody-bound state is aberrant. First, we showed that monobody binding has only a minor effect on transport function, and second, our model corresponds closely to the helix density in the EM dataset, which was obtained without exogenous binding proteins(Ubarretxena-Belandia et al., 2003). Although local perturbations at the monobody-binding interface of loops 1_A_ and 1_B_ cannot be ruled out, the position of loop 1_A_ is consistent with prior spectroscopic data, which indicated that in the major solution conformation, F27_A_ packs against the B subunit with its sidechain oriented towards the substrate binding site. Loop 1_B_ is located on the open side of the transporter and does not form any intra-transporter contacts. Therefore, even if monobody does stabilize a less-prevalent conformation of loop 1_B_, this would not change the major interpretations of the present structures.”

This was discussed in more detail in the prior Gdx-Clo structure paper from the Stockbridge group, but revisiting this is important here because of the role the loops play in closing off the transporter on one side of the membrane. Also, can this structure provide insight into how the L51I (and I62L) mutation near the end of TM2 (Lehninger eLife 2019;8:e48909) preferentially stabilizes EmrE in an open-to-one side conformation?

We have added the following paragraph to the discussion:

“Our results also provide some insight into the observation that a single L51I or I62L mutation in one subunit of the EmrE dimer prevents conformational exchange (Leninger et al., 2019). Both residues are located on transmembrane helices and are buried at protein interfaces in one monomer and accessible in the other (L51 to the aqueous binding pocket and I62 to the membrane). For Gdx-Clo, we previously posited that differential packing of the two monomers in the N-terminal half of helix 3 contributes to structural frustration and the resulting conformational exchange (Kermani et al., 2020). In EmrE, I62 is located in this same crucial region, and its mutation in only one monomer presumably disturbs the well-matched competition that occurs in the homodimer.*”*

2. What is the exact pH at which each of the substrate-bound crystal structures were determined? The methods state that the substrate-bound structures (drug-like-substrate, proton is also a substrate) were crystallized at pH 6.5 or pH between 7.1-7.3 depending on the buffer. But we do not see the exact pH listed in the tables for each of the different crystal forms or discussed in the text. This information is important because of the proton-coupled transport mechanism.The results state "To understand how different substrates interact with EmrE, we screened a variety of transported compounds in crystallization trials at pH values {greater than or equal to}6.5, where the E14 sidechains are expected to be deprotonated, favoring binding of the positively charged substrates." This is incorrect.

We have added the crystallization conditions for each structure to the data collection and refinement table (Table 1).

We have removed this sentence from the results. In the discussion, we elaborated on the interplay of pH and substrate binding as follows:

“In addition to a substrate-free, pH 5.2 structure, we solved structures of EmrE with methyl viologen, harmane, Me-TPP^+^, TPP^+^, and benzyltrimethylammonium at pH values between 6.3 and 7.5. Experiments with EmrE in bicelles have suggested that a proton can bind simultaneously with TPP^+^ with a pK_a_ of 6.8 (Robinson et al., 2017). In the NMR model, under conditions that favor simultaneous substrate and proton binding, F-TPP^+^ is positioned higher in the binding pocket, 2 Å closer to E14_B_ than protonated E14_A_ (Shcherbakov et al., 2021). In contrast, in our TPP^+^-bound structure, which was obtained at a pH of 7.25, TPP^+^ is situated lower in the binding pocket and within 0.5 Å of the midpoint between the glutamates. It is thus probable that this crystal structure represents the doubly-deprotonated, substrate-bound state. It is also likely that both glutamates are deprotonated in the methyl viologen-bound structure, since this substrate bears a +2 charge, making glutamate protonation more electrostatically unfavorable than in the presence of a monovalent substrate.

Protonation of the central glutamates has not been evaluated in the presence of monovalent substrates other than TPP^+^, and the E14 pK_a_ values are likely to vary according to factors such as binding pocket solvation or charge delocalization on the substrate. For the Me-TPP^+^, harmane, and benzyltrimethylammonium-bound structures (pH 6.5, 7.1, and 7.25, respectively), the contribution of a substrate+proton-bound population cannot be ruled out. However, the positioning of each of these substrates centered close to the midpoint between the E14 carboxylate groups, similar to TPP^+^, implies that in the major component of the population, both glutamates bear a negative charge.”